# AUTOMETRICS: APPROXIMATE HUMAN JUDGMENTS WITH AUTOMATICALLY GENERATED EVALUATORS

**Michael J. Ryan♠, Yanzhe Zhang♠, Amol Salunkhe♣, Yi Chu♣, Di Xu♣, Diyi Yang♠**
♠Stanford University    ♣American Express
michaeljryan@stanford.edu

## ABSTRACT

Evaluating user-facing AI applications remains a central challenge, especially in open-ended domains such as travel planning, clinical note generation, or dialogue. The gold standard is user feedback (e.g., thumbs up/down) or behavioral signals (e.g., retention), but these are often scarce in prototypes and research projects, or too-slow to use for system optimization. We present **AutoMetrics**, a framework for synthesizing evaluation metrics under low-data constraints. AutoMetrics combines retrieval from **MetricBank**, a collection of 48 metrics we curate, with automatically generated LLM-as-a-Judge criteria informed by lightweight human feedback. These metrics are composed via regression to maximize correlation with human signal. AutoMetrics takes you from expensive measures to interpretable automatic metrics. Across 5 diverse tasks, AutoMetrics improves Kendall correlation with human ratings by up to 33.4% over LLM-as-a-Judge while requiring fewer than 100 feedback points. We show that AutoMetrics can be used as a proxy reward to equal effect as a verifiable reward. We release the full AutoMetrics toolkit and MetricBank to accelerate adaptive evaluation of LLM applications.

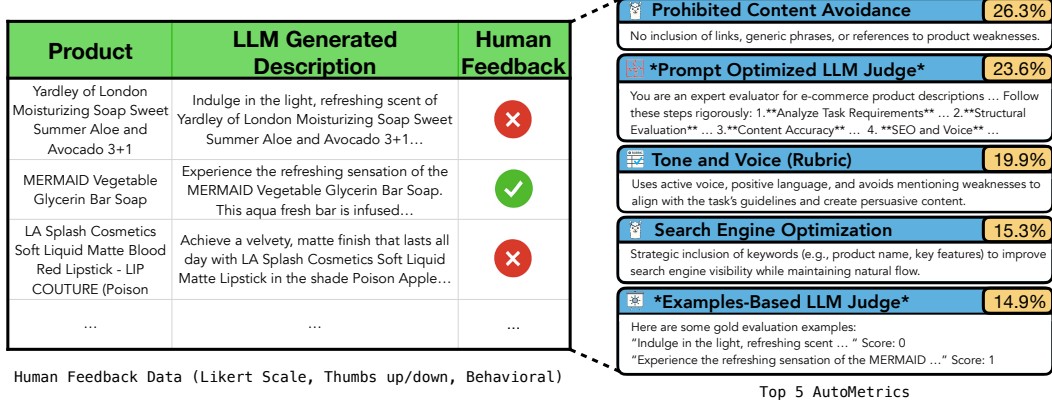

Figure 1: AutoMetrics takes you from expensive measures to interpretable automatic metrics. Here AutoMetrics generates useful metrics for evaluating LLM written product descriptions from user reviews from EvalGen (Shankar et al., 2024b). Percentages indicate relative importance of each metric derived from regression coefficients.

## 1 INTRODUCTION

Modern AI systems now demonstrate massively multitask capabilities imparted through extensive pretraining (Radford et al., 2019; Brown et al., 2020). Practitioners can rapidly prototype new AI-enabled tasks – from travel planning to code completion – at a pace much faster than the community

---

This paper reflects the academic work of the authors and does not represent or constitute the views, policies, positions, or practices of American Express or its affiliates.

can craft domain specific metrics (Papineni et al., 2002; Lin, 2004; Xu et al., 2016). This new era, in which large language models can be adapted to virtually any domain, places mounting pressure on evaluation practices. A divide is growing between tasks with easily verifiable rewards, such as math (Glazer et al., 2024; Shao et al., 2024) and coding (Chen et al., 2021), while subjective and open-ended tasks such as writing (Gurung & Lapata, 2025) remain difficult to measure. For these tasks, human evaluation remains the gold standard (Shankar et al., 2024b; Chiang et al., 2024).

Unfortunately, human evaluation is costly, slow, and not scalable for every prototype or user population. Reward models offer an alternative (Mnih et al., 2015; Christiano et al., 2017), but they typically require thousands of labels. The common alternative is rubric-based LLM-as-a-Judge methods (Li et al., 2023; Zheng et al., 2023; Liu et al., 2024), which rely on the assumption that system behavior is clearly defined and are not guaranteed to follow given rubrics strictly (Tripathi et al., 2025). In reality, practitioners typically have access only to non-descriptive human signals (e.g., thumbs up/thumbs down collected from users). In this setting, the problem is not only formulating the rubric, but also discovering the underlying criteria that matter.

This highlights the need for **dynamic, task-specific metric learning**. Instead of relying exclusively on human judgment or fixed rubrics, evaluation itself must become adaptive. Current efforts have emphasized making LLMs better evaluators of task-specific criteria (Liu et al., 2024; Kim et al., 2025; Anugraha et al., 2025) or leveraging rubrics to optimize LLMs (Gunjal et al., 2025; Viswanathan et al., 2025) but comparatively little work has focused on automatically generating the rubrics and criteria to be adaptively aligned with human judgment (Biyani et al., 2024; Ryan et al., 2025; Dunlap et al., 2025). Such adaptive evaluation is essential not only for easily assessing new tasks but also for optimizing evaluated systems based on real-time user feedback.

We introduce **AutoMetrics**, a method for dynamic metric induction that turns sparse, non-descriptive human feedback into actionable and interpretable evaluators (Figure 1). Starting from a task description and fewer than 100 human signals, AutoMetrics synthesizes candidate criteria, retrieves and adapts existing metrics, and composes them through regression into predictive measures of quality. Beyond simply identifying criteria, **AutoMetrics grounds and weighs them**, producing metrics that are both predictive and interpretable. This approach achieves up to **33.4% higher Kendall correlation** with human judgments than LLM-as-a-Judge baselines (§4), is **data-efficient** only requiring ∼80 feedback points (§4.6), and even **matches verifiable rewards** when optimizing downstream AI systems (§5). Beyond accuracy, **AutoMetrics reveals actionable insights into what users value**. We release AutoMetrics as an open-source toolkit[1], offering the community a powerful new way to evaluate and optimize AI applications at the speed of modern development.

## 2 RELATED WORK

**Metric Collections** Prior work has organized collections of metrics primarily for the ease of use on the part of the practitioner. When already using a library such as PyTorch (Paszke et al., 2019) or Huggingface (Wolf et al., 2020) it's simple to utilize TorchMetrics (Nicki Skafte Detlefsen et al., 2022) or HuggingFace `lighteval` (Fourrier et al., 2023). Scikit Learn Metrics (Pedregosa et al., 2011) and NLTK metrics (Bird & Loper, 2004) were created with the same intentions. All text-generation metrics covered by these collections are also contained in our MetricBank collection. Beyond integrating with existing open source libraries, some metric collections are part of ML observability frameworks like Evidently (EvidentlyAI, 2025), Galileo (Galileo, 2025), Scorecard.io (Doe & Devireddy, 2024), and DeepEval (ConfidentAI, 2025). Most metrics are tightly coupled with their observability platform, although Evidently and DeepEval offer open-source versions. While DeepEval offers a metric recommendation feature, it is based on a predefined decision tree of questions like "*Does your LLM application use Retrieval-Augmented Generation (RAG)?*" and "*Is LLM safety a priority for you?*". Most similar to our work is the MetaMetrics collection (Winata et al., 2025), which computes a regression over multiple task-specific metrics for tasks like image captioning and summarization to select the best combination of metrics. We compare our approach with MetaMetrics in Section 4 and find that our core thesis of adaptive metric generation is critical for evaluation in the low-data, novel task settings of interest.

---

[1]https://github.com/SALT-NLP/autometrics

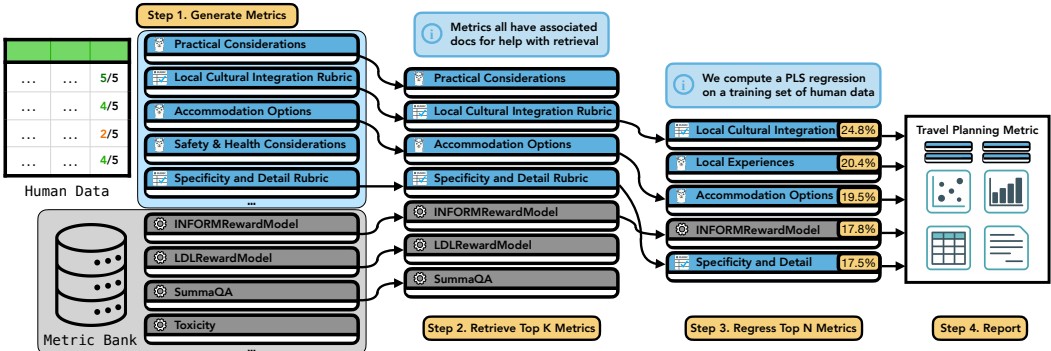

Figure 2: **AutoMetrics** comprises four steps. (1) *Generate*: create task-specific candidate metrics (Single criteria, Rubric, Examples, MIPROv2). (2) *Retrieve*: from the generated candidates plus MetricBank, use ColBERT to prefilter to $k'$ metric cards and an LLM to select the final $k$. (3) *Regress*: fit a PLS model on the training set to weight and select metrics that predict human judgments. (4) *Report*: produce a writeup with weights and correlations and details to guide adoption.

**LLM Based Evaluation**   LLM-as-a-Judge (Zheng et al., 2023) evaluation is increasingly popular with the frequent improvement of LLM capabilities. Several works devise task-specific prompts to enable LLM-based evaluation for storytelling (Chiang & Lee, 2023), summarization (Wang et al., 2023; Hada et al., 2024; Wu et al., 2023), dialogue (Lin & Chen, 2023; Fu et al., 2024), knowledge (Bai et al., 2023), translation (Kocmi & Federmann, 2023), and more (Brake & Schaaf, 2024). Another promising direction is devising frameworks and general methods for making LLM-as-a-Judge more reliable. G-Eval (Liu et al., 2023) proposes breaking LLM evaluation into a step-by-step chain of thought and taking a weighted sum over the log probabilities of generating different scores. Chat-Eval (Chan et al., 2024) simulates multiple perspectives by evaluating through multi-agent debate. SPADE (Shankar et al., 2024a) generates assertions for LLMs to verify based on labeled good and bad examples. VERDICT (Kalra & Tang, 2025) introduces judge-time scaling by decomposing judgments into composable units of reasoning, verification, debate, and aggregation steps. Though we take inspiration from many of these frameworks, the most directly similar to our LLM-as-a-Judge steps in the AutoMetrics pipeline are DnA-Eval (Li et al., 2025) and EvalGen (Shankar et al., 2024b). DnA-Eval (Li et al., 2025) decomposes evaluation into rubric criteria and aggregates the results across the criteria. EvalGen (Shankar et al., 2024b) elicits limited human feedback on generated outputs, proposes criteria for evaluation based on this feedback, and iteratively refines the criteria with a human-in-the-loop and LLM.

## 3   THE AUTOMETRICS METHOD

The purpose of AutoMetrics is to produce metrics for subjective and novel AI-enabled tasks. Our goal is to induce metrics that correlate strongly with human judgments while requiring minimal data collection. To accomplish this, we present a general pipeline with four stages: (1) generate, (2) retrieve, (3) regress, and (4) report. These steps are visualized in Figure 2. Each stage involves design choices among several alternatives, which we empirically validate (§4.5).

### 3.1   METRIC PRODUCTION

**Generate**   For sufficiently novel settings, generating criteria for LLM-as-a-Judge evaluation is essential. Broad coverage of evaluation criteria allows us to later filter down to what matters most. Accordingly, our default configuration generates **10 Single Criterion** LLM Judge metrics, **5 Rubric** LLM-Judge metrics, **1 Example**-based optimized LLM-Judge metric (fewshot), and **1 Prompt-Optimized** LLM-Judge metric per run of AutoMetrics[2]. Optimized metrics require more LLM calls/tokens to produce, while criteria and rubrics are relatively inexpensive. Unless otherwise specified, we use this configuration throughout the paper. Empirically, we find this mix of generated metrics

---

[2]Design details and ablations are in Appendix E.2; we validate these choices across nearly 30 settings.

generalizes across diverse domains and tasks. For each metric, we also generate a Metric Card documenting its description, intended use, implementation details, and limitations (Appendix B).

**Retrieve** In addition to generated metrics, we leverage our MetricBank: a collection of 48 metrics (Appendix Table 4) drawn from the NLP literature, each implemented and documented with a Metric Card. Directly evaluating all metrics would be prohibitively expensive, so we instead use retrieval as a filtering step. We treat Metric Cards as documents, and use a description of the evaluation setting or task as the search query. Retrieval is performed using a hybrid **ColBERT + LLM** approach,[3] narrowing the candidate pool to metrics most relevant to the task at hand.

**Regress** The filtered pool of candidate metrics must still be combined into a predictive signal for human judgment. We normalize all metric scores to their z-scores and fit a **Partial Least Squares (PLS)** regression model. Intuitively, PLS projects the metric space onto the direction most predictive of human labels, then regresses labels along that axis. We choose PLS regression because it works well under the constraints of our setting that: (1) the number of predictors (metrics) may be comparable to or larger than the number of observations (data points), and (2) the predictors are often highly correlated. Concretely, with a single latent component, PLS finds a unit weight vector $w^\star \in \mathbb{R}^d$ that maximizes

$$w^\star \; = \; \arg \max_{\|w\|_2 = 1} \; \mathrm{cov}(Xw, \, y)^2,$$

where $X$ is the matrix of normalized metric scores and $y$ is the vector of human labels. The latent score is $t = Xw^\star$, and PLS then regresses the human labels on this latent score, yielding predictions $\hat{y} = t\beta$ with coefficient $\beta = \frac{t^\top y}{t^\top t}$.

We apply this procedure in two stages. In the first stage, we fit PLS using all candidate metrics and rank them by the magnitude of their weights in $w^\star$. We then select the top $n$ metrics according to this ranking. In the second stage, we refit PLS on this reduced set of $n$ metrics to obtain a new projection $t$ and corresponding predictions $\hat{y}$. As a final step, we remove negatively correlated LLM-generated metrics, as they are designed to target positive correlation. We don't apply this to existing measures (e.g., length can negatively correlate with conciseness).

### 3.2 METRIC EVALUATION

To evaluate the quality of induced metrics, we draw on concepts of measurement validity from research (Borsboom et al., 2004) and testing (American Educational Research Association et al., 2014). We focus on three forms: "Content Validity", "Criterion Validity", and "Construct Validity".

**Content Validity** asks whether a metric represents the construct it is intended to measure. Although direct quantification is difficult, we encourage transparency by releasing metric reports. Because our generated metrics rely on LLM judges, we also expose the reasoning traces of the judge LLM, allowing users to inspect whether assessments appear justified. These traces can further aid system optimization with AutoMetrics (§5).

**Criterion Validity** Criterion validity measures correlation with a reference standard. In NLP, correlation with human labels has been the most widely used criterion (Banerjee & Lavie, 2005; Xu et al., 2016; Gehrmann et al., 2021). We assess criterion validity by comparing AutoMetrics to ground-truth human labels. We report Kendall's $\tau$, which makes no distributional assumptions and simply checks whether the rank order induced by a metric matches that of human judgments. This provides a conservative estimate compared to Spearman's $\rho$ or Pearson's $r$.

**Construct Validity** measures whether a metric captures an underlying abstract concept, such as "quality." Both human judgments and AutoMetrics attempt to approximate "quality". We draw from convergent–discriminant validity (Campbell & Fiske, 1959) and operationalize construct validity as robustness. A useful metric should penalize quality degradations (sensitivity) while remaining stable under equivalent-quality variation. In order to quantify convergent-discriminant validity, we introduce two measurements: **Sensitivity** and **Stability**. To construct test cases, we use an LLM to generate strategies for degrading outputs on a given dataset, and apply these to produce *worse-quality perturbations*. In contrast, *same-quality perturbations* are produced from a fixed set of hand-crafted

---

[3]We ablate the selection algorithm in Appendix E.1.

transformations—such as rephrasing, reordering, synonym replacement, or stylistic edits—that are designed to preserve the target evaluation dimension. Prompts are provided in Appendix C.

- **Sensitivity** measures whether a metric assigns lower scores to degraded outputs. Let $s_{\text{orig}}^{(i)}$ and $s_{\text{worse}}^{(i)}$ denote the normalized scores for the original and worse-quality perturbed outputs of sample $i$ from a dataset of size $|N|$. Sensitivity is defined as:

$$\text{Sensitivity} = \frac{1}{N} \sum_{i=1}^{N} \mathbf{1}\left[ s_{\text{worse}}^{(i)} < s_{\text{orig}}^{(i)} \right]$$

- **Stability** measures whether a metric produces consistent scores when quality should be preserved. Let $s_{\text{same}}^{(i)}$ be the normalized score for a same-quality perturbation of sample $i$ from a dataset of size $|N|$. Stability is defined as:

$$\text{Stability} = 1 - \frac{1}{N} \sum_{i=1}^{N} \left| s_{\text{orig}}^{(i)} - s_{\text{same}}^{(i)} \right|.$$

High sensitivity indicates strong penalization of degraded outputs, while high stability indicates invariance to irrelevant variation. Both are desirable, and together they provide a general-purpose lens for evaluating how well a metric generalizes.

## 4 EXPERIMENTS AND EVALUATIONS: SHOWING AUTOMETRICS ARE VALID

For our experiments, we focus on showing that our AutoMetrics are valid across many tasks/domains and that they correlate better with human judgements than competitive baselines. We showcase AutoMetrics have high Criterion Validity and Construct Validity across several tasks.

### 4.1 TASKS

| Dataset (Citation) | Task | Domain | # Data | Feedback | # Eval Dim | Refs |
|---|---|---|---|---|---|---|
| ***In-Distribution Tasks***: *some metrics in our bank were designed to directly evaluate these tasks.* | | | | | | |
| SimpEval (Maddela et al., 2023) | Simplification | 📖 | 360 | 1–100 Likert | 1 | ✓ |
| HelpSteer2 (Wang et al., 2024) | Dialogue | 💬 | 20,324 | 1–5 Likert | 5 | ✗ |
| ***Out-of-Distribution Tasks***: *no metric is specifically designed for these – tests generalization and metric generation.* | | | | | | |
| EvalGen (Shankar et al., 2024b) | Product description | 📄 | 100 | Binary | 1 | ✗ |
| RealHumanEval (Mozannar et al., 2025) | Code completion | ⟨/⟩ | 5,204 | Behavioral | 1 | ✗ |
| Co-Gym (Shao et al., 2025) | Travel planning | ✈ | 72 | 1–5 Likert | 3 | ✗ |

Table 1: Overview of tasks. **Icons:** ⟨/⟩ Code Generation; 📄 Data-to-Text Generation; 💬 Dialogue/Chat; 📖 Education/Readability; ✈ Travel Planning.

In order to evaluate our AutoMetrics method, we collect two types of tasks: *In-Distribution Tasks*, which are tasks where some of the metrics in our Metric Bank were designed to directly evaluate the task, and *Out-of-Distribution Tasks*, which are tasks where no metric in particular was designed to assess the task. All of our tasks utilize human feedback for evaluation, encompassing behavioral feedback, binary feedback (thumbs up/down), and Likert scale feedback, which is already collected as part of the dataset. We introduce all tasks in Table 1. In our main tables we present results for five datasets and a single evaluation dimension from each: **SimpEval** (Maddela et al., 2023) (sentence simplification score 1–100), **HelpSteer2** (Wang et al., 2024) (Chatbot helpfulness 1–5), **EvalGen** (Shankar et al., 2024b) (Product Review Thumbs Up/Down), **RealHumanEval** (Mozannar et al., 2025) (accepted or rejected code edit), **CoGym** (Shao et al., 2025) (travel plan outcome rating 1–5). We report evaluations on more settings in the Appendix results.

### 4.2 BASELINES

We include the following baselines: **Best Existing Metric**, where we run all 48 metrics (or 19 metrics for reference-free tasks), record their Kendall correlation on the validation set, and select

the best metric to use for the task based on the validation correlation. **MetaMetrics**, where we take all the metrics from the MetaMetrics paper and compute an XGBoost Regression on the metrics on the trainset (Winata et al., 2025). **Finetuned LLM** refers to training a `ModernBERT-large` (Warner et al., 2024) to predict the human annotation. We implement it by training LoRA adapters (Hu et al., 2021) with rank $= 16$ on all the attention, dense layers, and regression head, using a learning rate of $5e - 5$ and a batch size of 16 for three epochs over the training data. For the **LLM-Judge** baseline, we use the original human annotation prompt for each task and provide it to an LLM. We include all of these prompts in Appendix C. **DnA-Eval** (Li et al., 2025) involves using an LLM to generate three dimensions where a user request may benefit from evaluation, along with weights for how to aggregate these dimensions. Then each of those dimensions is scored with an LLM-as-a-Judge, and finally aggregated based on the LLM-generated weights.

## 4.3 CRITERION VALIDITY (CORRELATION)

We report Kendall's $\tau$ of all methods with GPT-4o-mini and Qwen-3-32B Reasoning in Table 2.

| Method | In-Distribution | | Out-of-Distribution | | |
| | SimpEval | HelpSteer2 | EvalGen | RealHumanEval | CoGym |
| --- | --- | --- | --- | --- | --- |
| **Model Agnostic** | | | | | |
| Best Existing Metric | $0.246 \pm 0.00$ | $0.327 \pm 0.00$ | $0.193 \pm 0.00$ | $0.138 \pm 0.00$ | $0.074 \pm 0.00$ |
| MetaMetrics (Winata et al., 2025) | $0.127 \pm 0.01$ | $0.204 \pm 0.00$ | $-0.214 \pm 0.01$ | $0.025 \pm 0.01$ | $-0.119 \pm 0.02$ |
| Finetuned LLM | $0.076 \pm 0.08$ | $0.039 \pm 0.03$ | $0.054 \pm 0.05$ | $0.049 \pm 0.06$ | $0.223 \pm 0.20$ |
| **GPT-4o-mini Backbone** | | | | | |
| LLM-Judge | $0.272 \pm 0.02$ | $0.259 \pm 0.01$ | $0.161 \pm 0.14$ | $0.069 \pm 0.01$ | $\mathbf{0.199} \pm 0.13$ |
| DnA Eval (Li et al., 2025) | $0.234 \pm 0.03$ | $0.255 \pm 0.02$ | $0.174 \pm 0.16$ | $0.152 \pm 0.01$ | $0.185 \pm 0.10$ |
| AutoMetrics (Ours) | $\mathbf{0.321} \pm 0.04$ | $\mathbf{0.324} \pm 0.01$ | $\mathbf{0.334} \pm 0.06$ | $\mathbf{0.160} \pm 0.00$ | $-0.034 \pm 0.17$ |
| **Qwen-3-32B Backbone** | | | | | |
| LLM-Judge | $0.294 \pm 0.04$ | $0.334 \pm 0.02$ | $0.272 \pm 0.13$ | $0.025 \pm 0.01$ | $0.276 \pm 0.19$ |
| DnA Eval (Li et al., 2025) | $0.042 \pm 0.04$ | $0.260 \pm 0.02$ | $0.232 \pm 0.19$ | $0.071 \pm 0.15$ | $0.353 \pm 0.25$ |
| AutoMetrics (Ours) | $\mathbf{0.316} \pm 0.02$ | $\mathbf{0.342} \pm 0.01$ | $\mathbf{0.382} \pm 0.05$ | $\mathbf{0.145} \pm 0.00$ | $\mathbf{0.365} \pm 0.08$ |

Table 2: Criterion Validity results showing Kendall's Tau with 95% confidence intervals over 5 independent runs. AutoMetrics outperforms the baselines on all five tasks with Qwen3-32B and is within 95% confidence of the best for 4/5 tasks with GPT-4o-mini. On EvalGen, AutoMetrics improves performance by 33.4% over the closest baseline (LLM Judge).

**AutoMetrics correlates better than all baselines across all five tasks.** We find that AutoMetrics outperforms all other existing baselines on all five tasks. While the best performing baseline is both inconsistent on dataset (LLM Judge on SimpEval, HelpSteer, EvalGen; DnA Eval on RealHumanEval and CoGym) and on the underlying model used (Existing Metrics outperform GPT-4o-mini but not Qwen3-32B). In contrast, AutoMetrics is consistently the best option regardless of dataset or underlying model. On all datasets besides HelpSteer and CoGym, the AutoMetrics performance exceeds all baselines by greater than the 95% confidence interval. In general, AutoMetrics is the best choice for higher correlation with human ratings.

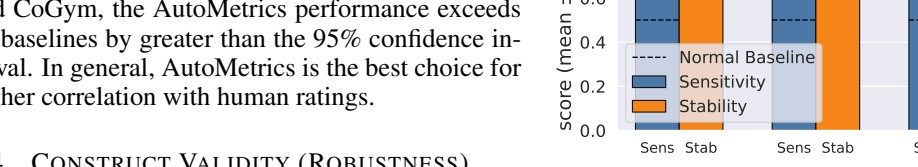

Figure 3: Sensitivity/Stability of AutoMetrics for SimpEval, HelpSteer2, and CoGym. AutoMetrics are sensitive to negative perturbations and stable on neutral perturbations.

## 4.4 CONSTRUCT VALIDITY (ROBUSTNESS)

To measure construct validity, we take inspiration from convergent-discriminant validity and show that AutoMetrics are strong predictors when output quality degrades and that they are stable under unimportant perturbations. To do so we introduced **Sensitivity** and **Stability** (§3.2). Sensitivity measures the rate of detection of negative perturbations and

Stability measures the magnitude of score preservation under meaningless changes. We report Sensitivity and Stability for all metrics on 30 trials in Figure 3. We compare against a normal distribution baseline.

**AutoMetrics are sensitive and stable.**   AutoMetrics are sensitive to degradation in output quality in 81.0-97.8% of cases, depending on the dataset, which is significantly greater than the 50% baseline. AutoMetrics can be a strong tool for identifying degradations in output quality. Similarly, AutoMetrics also always outperforms the baseline for stability by greater than 95% confidence intervals. Under insignificant modifications to evaluated outputs, AutoMetrics are consistently stable.

## 4.5   DESIGN DECISIONS (HYPERPARAMETER SWEEPS)

Our sweeps/ablations test three parts of the AutoMetrics method: the MetricBank, the retrieval step, and the regression step. We report Kendall's $\tau$ rank correlation across our six main tasks with 95% confidence intervals over five runs in Table 3. All sweeps and ablations are instead done on the dev set for all datasets. We never make design decisions based on runs of our test sets.

| | **In-Distribution** | | **Out-of-Distribution** | | |
| Method | SimpEval | HelpSteer2 | EvalGen | RealHumanEval | CoGym |
|---|---|---|---|---|---|
| **MetricBank Ablations (k=30; n=5)** | | | | | |
| Existing Metrics Only | $0.238 \pm 0.04$ | $\underline{0.376} \pm 0.00$ | $0.389 \pm 0.00$ | $\mathbf{0.155} \pm \mathbf{0.00}$ | $0.258 \pm 0.00$ |
| Generated Metrics Only | $\mathbf{0.276} \pm \mathbf{0.03}$ | $0.308 \pm 0.01$ | $\mathbf{0.503} \pm \mathbf{0.03}$ | $0.132 \pm 0.00$ | $\mathbf{0.433} \pm \mathbf{0.04}$ |
| Full MetricBank | $\underline{0.275} \pm 0.02$ | $\mathbf{0.387} \pm \mathbf{0.00}$ | $0.474 \pm 0.03$ | $\underline{0.152} \pm 0.01$ | $0.329 \pm 0.02$ |
| **Retrieval Ablations (n=5)** | | | | | |
| Retrieve k=5 | $0.257 \pm 0.03$ | $0.336 \pm 0.03$ | $0.414 \pm 0.12$ | $0.124 \pm 0.02$ | $\mathbf{0.385} \pm \mathbf{0.04}$ |
| Retrieve k=10 | $0.245 \pm 0.02$ | $0.352 \pm 0.01$ | $0.469 \pm 0.06$ | $0.128 \pm 0.01$ | $\underline{0.371} \pm 0.02$ |
| No Metric Cards (k=20) | $\underline{0.281} \pm 0.04$ | $0.328 \pm 0.02$ | $0.427 \pm 0.09$ | $0.134 \pm 0.01$ | $0.292 \pm 0.06$ |
| Retrieve k=20 | $\mathbf{0.286} \pm \mathbf{0.02}$ | $\underline{0.378} \pm 0.01$ | $\mathbf{0.522} \pm \mathbf{0.02}$ | $0.141 \pm 0.01$ | $0.302 \pm 0.06$ |
| Retrieve k=30 | $0.275 \pm 0.02$ | $\mathbf{0.387} \pm \mathbf{0.00}$ | $\underline{0.474} \pm 0.03$ | $\mathbf{0.152} \pm \mathbf{0.01}$ | $0.329 \pm 0.02$ |
| **Regression Ablations (k=30)** | | | | | |
| No Regression (n=1) | $0.232 \pm 0.08$ | $\mathbf{0.393} \pm \mathbf{0.00}$ | $0.353 \pm 0.23$ | $0.145 \pm 0.00$ | $0.356 \pm 0.00$ |
| Regress n=3 | $0.255 \pm 0.02$ | $\underline{0.389} \pm 0.02$ | $\mathbf{0.503} \pm \mathbf{0.10}$ | $\underline{0.152} \pm 0.01$ | $0.302 \pm 0.04$ |
| Regress n=5 | $\underline{0.275} \pm 0.02$ | $0.387 \pm 0.00$ | $0.474 \pm 0.03$ | $\underline{0.152} \pm 0.01$ | $0.329 \pm 0.02$ |
| Regress n=10 | $\mathbf{0.309} \pm \mathbf{0.01}$ | $0.358 \pm 0.01$ | $0.461 \pm 0.05$ | $0.147 \pm 0.01$ | $0.297 \pm 0.05$ |
| Regress n=20 | $0.268 \pm 0.03$ | $0.350 \pm 0.01$ | $\underline{0.498} \pm 0.04$ | $\mathbf{0.153} \pm \mathbf{0.01}$ | $\mathbf{0.361} \pm \mathbf{0.02}$ |

Table 3: Kendall correlation with 95% confidence intervals on in-distribution and out-of-distribution datasets over five runs with Qwen3 32B (Reasoning). The Full MetricBank and Metric Cards prove useful, and the best settings for retrieval and regression are k=30 and n=5 respectively.

**Both Generated and Existing Metrics Help.**   In all of our tasks, the Full MetricBank was either the best or second-best performing setting for the ablations. When it was second best, it was typically within 95% confidence intervals. The primary exception is CoGym, where "Full MetricBank" fell 0.104 below "Generated Metrics Only" and, to a lesser extent, EvalGen, where "Full MetricBank" was short by 0.029. CoGym and EvalGen are also our smallest training sets (37 and 57 training samples respectively). We hypothesize this is because on out-of-distribution tasks, existing metrics tend to be noisy predictors which can spuriously correlate during the regression. Generated metrics tend to be less noisy predictors. Larger training sets provide a more effective filter for identifying useful metrics. We further explore this hypothesis in our data scaling experiment (§4.6).

**Metric Cards Help Retrieval and Larger k Is Better.**   Across all five tasks, retrieval with Metric Cards (k=20) is better than retrieval without metric cards (using a single sentence description of the metric). Furthermore, we see roughly linear growth of correlation with higher $k$ metrics retrieved to run on the train set. The single exception to this trend is CoGym, which can be attributed to the noisiness of the small dataset and the generated metrics being less noisy predictors. The top 5 retrieved metrics are often generated ones, reducing the risk of recommending spuriously correlated existing metric on the small dataset. We ran all retrieval experiments by regressing with $n = 5$, so

it is worth noting that future improvements to the retrieval algorithm (possibly including historical usage data) mean that it is feasible for $k = 5$ numbers to match our $k = 30$ results, so long as the proper metrics are recommended.

**Number to regress to varies from dataset to dataset, but five is a good average case.** The best case for regression only repeats once (with n=20), suggesting that the number of metrics needed is highly dependent on the complexity of the evaluation task and domain. Since there is no clear winner, we select n=5 as a default because it is the second best in two of five tasks, and it is the cheapest option that still maintains lower variance from run to run. A higher N means producing more expensive metrics to run downstream, so n=5 is a useful compromise of cost and performance.

### 4.6   HOW MUCH DATA DO YOU NEED TO USE AUTOMETRICS?

To test how much data is needed to use AutoMetrics, we test on three distinct datasets large enough to be useful in this experiment. We take a relatively simple *In-Distribution* dataset, SimpEval, a more challenging *In-Distribution* dataset, HelpSteer2, and an *Out-of-Distribution* dataset RealHumanEval. We vary the train set size from N=5, 10, 20, 40, 80, 160, and (for RealHumanEval and Helpsteer2) 320 and 640. We run these settings for both the "Generated Only" Metric Bank and "Full" Metric Bank (with existing metrics). We plot the correlation on the full test set in Figure 4.

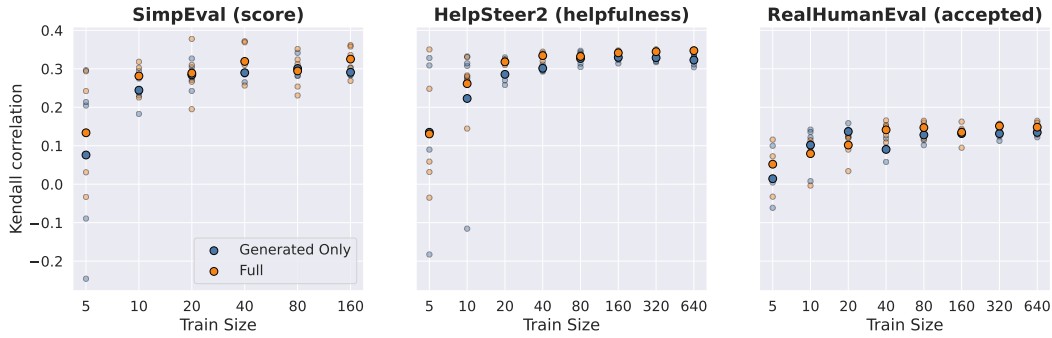

Figure 4: All correlations plotted for various training set sizes with "Generated Only" and "Full" Metric Banks. Individual trials are translucent while average performance at a scale is solid.

**About 80 samples saturates performance.** Across all three datasets and both settings, performance levels off after about 80 samples. It is possible with more sophisticated metric generation/learning methods more data could continue to help, however with the current architecture between 80-100 examples is all you need. Below 80 examples most of the lower performance is due to the high variance of fitting a regression to a small training set.

**On out-of-distribution datasets "Generated Only" can outperform "Full" with low-resources.** Looking to the RealHumanEval plot we see at training size 10 and 20 the "Generated Only" metrics outperform the "Full" Bank. Recall back to the ablations (§4.5) where we observed on the small, out-of-distribution datasets, CoGym and EvalGen, that "Generated Only" outperformed the "Full" MetricBank. Since most tasks will be out of distribution by nature, we default to using "Generated Only" when the user provides less than 80 training samples. Beyond 80, both "Generated Only" and "Full" level off, however "Full" asymptotes higher than "Generated Only" on all datasets. We argue this is a product of the high-p, low-n problem in regression where having too many weak predictors and not enough datapoints can lead to spurious correlations. By limiting to generated metrics for low-n settings we enforce the use of stronger predictor signals.

## 5   CASE STUDY: AUTOMETRICS FOR OPTIMIZING AN AGENTIC TASK

A natural extension to using AutoMetrics is to take the limited data one has available in order to learn a useful set of metrics that can then be used for optimizing a system. In this way AutoMetrics

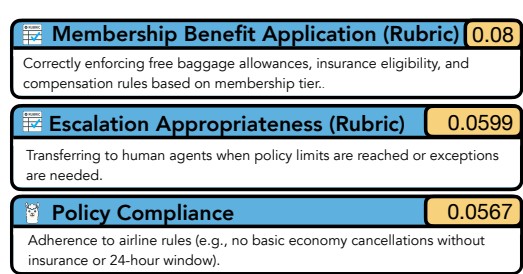

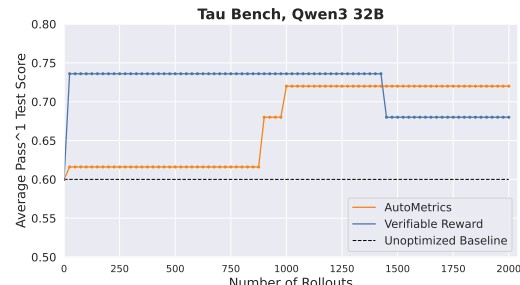

Figure 5: AutoMetrics produces three metrics for $\tau$-Bench. Regression coefficients in yellow.

Figure 6: $\tau$-Bench performance over GEPA optimization steps when using AutoMetrics.

would operate similar to the purpose of a Reward Model or a Verifiable Reward. In order to test if AutoMetrics can be useful in this setting we optimize an airline assistance agent for $\tau$-bench (Yao et al., 2024), a testbed for tool-use agents to interact with simulated users to accomplish tasks. We split the 50 $\tau$-airline tasks into 25 for training and 25 for evaluation.

**Simulating a verifiable reward.** To run AutoMetrics we rollout the 25 training examples 8 times each with temperatures [0.0, 0.01, 0.02, 0.03, 0.05, 0.1, 0.15, 0.2]. Then we obtain the true reward signal for each of these rollouts. In practice rather than a verifiable reward this could be a subjective human label. We run AutoMetrics in "Generate Only" mode and allocate more resources to generated metrics (10→20 llm judge metrics; 5→8 rubric metrics). Otherwise we run with default hyperparameters (k=30; n=5). We show the generated metrics in Figure 5.

AutoMetrics recommends three metrics for Tau-Bench evals: two rubric based metrics and one single criterion metric. Originally our (n=5) setting recommended five metrics, however our final filtering step removed two metrics for having negative coefficients. Since the trajectories are only derived from 25 examples it is likely that metrics will begin to learn things about the data itself. This reflects the importance of both human oversight and our metric filtering.

**Optimizing without a Verifiable Reward** We implement a simple ReAct (Yao et al., 2023) agent in DSPy (Khattab et al., 2024) for performing the $\tau$-Airline task. Our baseline agent gets 60% accuracy on the 25 test examples averaged over five trials. We then run a baseline optimization where we use the DSPy GEPA optimizer (Agrawal et al., 2025) to optimize an agent on the 25 training tasks with **Verifiable Reward**. Next we run optimization with our **AutoMetrics** as the metric for GEPA optimization. We show the performance on the test set after N rollouts in Figure 6. We find that **AutoMetrics can match performance of a verifiable reward.** After 2000 rollouts the GEPA optimization with verifiable reward achieves $0.680 \pm 0.11$ accuracy over 5 trials while the AutoMetrics run gets $0.720 \pm 0.06$. AutoMetrics statistically significantly exceeds the baseline performance ($p < 0.05$) of 0.6. This demonstrates that AutoMetrics can match or exceed Verifiable Rewards as optimization signal.

## 6   DISCUSSION AND CONCLUSION

In this paper, we introduced **AutoMetrics**, a method for producing metrics that correlate with human judgments on subjective tasks. Requiring only ~80 human-labeled examples, AutoMetrics achieve high criterion validity (§4.3) and construct validity. (§4.4). AutoMetrics improve upon existing baselines by up to 33.4% in Kendall correlation with human ratings. In a case study on Tau-Bench, AutoMetrics matched or exceeded gains obtained from optimizing on a verifiable reward (§5).

We draw two key lessons for practitioners. First, **data diversity is critical**: while only ~80 feedback points suffice for moderate correlation (§4.6), scaling up synthetic data from limited sources can produce metrics that reflect dataset artifacts rather than system quality (§5). Second, **human oversight remains essential**: domain experts can help remove spuriously correlated metrics which the automatic filtering process misses. When using metrics for optimization, practitioners should monitor metric feedback and improvement with observability tools (Chavez, 2025).

Overall, **AutoMetrics** provides a practical first step for exploring data and guiding optimization when collecting preliminary human evaluation in new domains. The metrics it produces are interpretable, actionable, and informative for system improvement. We release AutoMetrics publicly and invite community contributions of new metrics and methods to strengthen the framework.

## REPRODUCIBILITY STATEMENT

AutoMetrics is intended to be an open source library and framework. As such we take great effort to make the running and evaluation of AutoMetrics user-friendly. We have attached an anonymized repository for AutoMetrics with this submission. In addition to the core algorithm, the repository also contains the python scripts to reproduce all experimental results in this paper. All of our design decisions, hyperparameters, and ablations are rigorously documented throughout the paper across Section 4.5 and Appendix E. We provide system-specs needed to run the metrics in Table 5. We also share the exact prompts and DSPy signatures used in calling LLMs in Appendix C. For all main experimental results (e.g. Table 2 and Table 3) results are reported over five independent random seeds to ensure findings are robust and statistically significant.

## LIMITATIONS

As a part of the AutoMetrics framework we construct and optimize metrics with particular LLMs. Because the metric generation process involves optimizing to a particular model we have found that producing metrics with one model and running them with another reduces performance. This suggests that when better models are released it will be important to reoptimize automatic metrics using AutoMetrics rather than just swap out the underlying LLM.

AutoMetrics may only generalize as far as the provided data enables it. Collecting real, diverse human data is still an essential part of evaluation. The more representative and generalizable the input data is, the better and more general the AutoMetrics will be. Users should collect data that is representative of the opinions and population that they want their evaluation to cover.

AutoMetrics depends on running a regression for many predictors on a limited number of data points. Although we took this into account with the design of our Regression step, it is still possible to run into a high-P low-N regression problem that risks spurious correlations. To counteract accidental misuse of AutoMetrics leading to poor evaluation, we add warnings to the metric reports when the significance of the correlation with human judgments of the recommended metric is low ($p > 0.05$).

Finally, as a part of this work we do not conduct a formal user study to demonstrate the adoption of AutoMetrics among practitioners. We have collected positive feedback on the metrics through informal tests with AI developers. We hope that by releasing and open sourcing this library, we will have the opportunity to work with the community to test and improve AutoMetrics.

## ACKNOWLEDGEMENTS

The authors would like to thank Omar Khattab, William Held, Saurabh Shah, Aryaman Arora, Ken Liu, David Anugraha, Vishakh Padmakumar, Hao Zhu, Lakshya Agrawal, Yu Fei, Seungone Kim, and Jonathan Hilgart for their insightful comments and thoughts at various stages of the project. We would also like to thank SALT Lab and the Stanford NLP Group for their help with the review and revision of the manuscript. Finally, we thank Yijia Shao and Alex Spangher for testing and providing feedback on the AutoMetrics system. This work has been supported in part through the Stanford HAI Corporate Affiliate Program, with membership funding provided by American Express. This work was also supported by the Sloan Foundation, Schmidt Sciences, and a grant under ONR N00014-24-1-2532.

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

## A   LLM USAGE ACKNOWLEDGMENT

LLMs were used to rephrase and edit writing in the paper, after an entirely human-written first draft. LLMs were also used as coding assistants in writing code for this project. All code and writing edits produced by LLMs were rigorously verified by the first-author.

## B   INTRODUCING METRICBANK

**MetricBank**   As our first significant contribution, we curate MetricBank, a standardized collection of 48 commonly used metrics in NLP literature. We source the metrics from Schmidtova et al. (2024), which examined all papers from the *International Conference on Natural Language Generation* (INLG) 2023 and all papers in the *Generation* track presented at ACL 2023, totaling 110 papers. They collected a list of all the Natural Language Generation (NLG) metrics used in those works, which totaled 283 different automatic metrics grouped into 34 metric families. We sorted by the most popular and implemented the top metrics from the top 16 families (28 metrics). Then, for completeness, we also implemented any remaining NLG metrics in NLTK Bird & Loper (2004), PyTorch Paszke et al. (2019), Huggingface Lighteval Fourrier et al. (2023), and Metametrics Winata et al. (2025) for an additional 12 metrics. Finally, we source a few additional metrics from recent papers not covered in the 2024 survey. We provide individual justifications for these 8 metrics in Appendix B.1.

We provide interesting stats about our metrics in Table 4. In particular, we collect 29 reference-based metrics, such as BLEU Papineni et al. (2002), which require a gold reference output, and 19 reference-free metrics, such as FKGL Flesch (1943), which measure quality of text without comparison to a reference. Our metrics span 12 distinct domains. We implement each metric with a simple interface of a `calculate` method that takes in the generated text and produces a floating-point score and optionally text feedback.

**Metric Cards**   Inspired by Model Cards Mitchell et al. (2019) and Data Cards Pushkarna et al. (2022) we design Metric Cards for simple documentation and reporting of the intended usage of metrics. Our Metric Cards contains seven main sections. **Metric Details** contains the description of the metric as well as core details that are needed to use it, such as the range of outputs, if it's reference based, if an input is required, etc. **Intended Use** describes the domain/tasks where the metric should be used as well as recommendations for when and when not to use the metric. **Metric Implementation** links to reference implementations and provides guidance on practical matters about the metric such as it's efficiency and scalability. **Known Limitations** explains biases, misuse, and known failure cases of the metric. **Related Metrics** links to similar metrics to help when browsing for the right metric for your task. **Further Reading** points to papers, blogs, and tutorials covering the metric. Finally, **Metric Card Authors** makes it clear who wrote the metric card and if they used an AI assistance. We provide a complete example of a metric card for the common BLEU metric Papineni et al. (2002) in Appendix D. We also provide a prompt for using LLMs to write a first pass of a metric card in Appendix C.

### B.1   ADDITIONAL METRICS

Here we provide justifications for our additional metrics that we did not collect from the metric survey (Schmidtova et al., 2024), lighteval (Fourrier et al., 2023), torchmetrics (Nicki Skafte Detlefsen et al., 2022), etc.

**Reward Models.**   We choose some of the most performant reward models off the RewardBench leaderboard (Lambert et al., 2024) at development time. The three models we used are **INFORM-RewardModel** (llama 3.1 70b) (Minghao Yang, 2024), **LDLRewardModel** (Gemma 2 27B) (Chen et al., 2025), and **GRMRewardModel** (Llama 3.2 3B) (Yang et al., 2024). We also add in the Qwen2.5 7B **Process Reward Model** (Zhang et al., 2025).

| Metric (Citation) | Domain | GPU | Type | Sup. | Default LLM |
|---|---|---|---|---|---|
| *Reference-Based Metrics: rely on a gold reference for comparison.* | | | | | |
| Jaccard Distance Jaccard (1901) | 📄,⇄,✎ | ✗ | edit-distance | ✗ | N.A. |
| Hamming Distance Hamming (1950) | ✎ | ✗ | edit-distance | ✗ | N.A. |
| Levenshtein Distance Levenshtein (1966) | 🌐,📄,⇄,✎ | ✗ | edit-distance | ✗ | N.A. |
| Levenshtein Ratio Levenshtein (1966) | 🌐,📄,⇄,✎ | ✗ | edit-distance | ✗ | N.A. |
| Jaro Similarity Jaro (1989) | ✎ | ✗ | edit-distance | ✗ | N.A. |
| Jaro–Winkler Winkler (1990) | ✎ | ✗ | edit-distance | ✗ | N.A. |
| BLEU Papineni et al. (2002) | 🌐 | ✗ | n-gram overlap | ✗ | N.A. |
| NIST Doddington (2002) | 🌐,📄 | ✗ | n-gram overlap | ✗ | N.A. |
| ROUGE Lin (2004) | 📄,🌐,⇄,📋 | ✗ | n-gram overlap | ✗ | N.A. |
| METEOR Banerjee & Lavie (2005) | 🌐,📄,⇄,🖼 | ✗ | n-gram overlap | ✗ | N.A. |
| TER Snover et al. (2006) | 🌐 | ✗ | edit-distance | ✗ | N.A. |
| iBLEU Sun & Zhou (2012) | ⇄,📖,💬 | ✗ | n-gram overlap | ✗ | N.A. |
| CHRF++ Popović (2015) | 🌐,📄,⇄,📋 | ✗ | n-gram overlap | ✗ | N.A. |
| CIDEr Vedantam et al. (2015) | 🖼 | ✗ | n-gram overlap | ✗ | N.A. |
| GLEU Wu et al. (2016) | 🌐 | ✗ | n-gram overlap | ✗ | N.A. |
| SARI Xu et al. (2016) | 📄,📖 | ✗ | n-gram overlap | ✗ | N.A. |
| CharCut Lardilleux & Lepage (2017) | 🌐,✎ | ✗ | edit-distance | ✗ | N.A. |
| MoverScore Zhao et al. (2019) | 🌐,📄,🖼,📋 | ✓ | embedding sim | ✗ | BERT |
| PseudoPARENT Dhingra et al. (2019) | 📄,📋,⇄ | ✗ | n-gram overlap | ✗ | N.A. |
| BERTScore Zhang et al. (2020) | 🌐,📄,⇄,🖼 | ✓ | embedding sim | ✗ | RoBERTa-Large |
| BLEURT Sellam et al. (2020) | 🌐,📄,⇄,📋 | ✓ | LM regression | ✓ | BERT/RemBERT |
| BARTScore Yuan et al. (2021) | 🌐,📄,⇄,📋,💬 | ✓ | LM regression | ✓ | BART |
| InfoLM Colombo et al. (2021) | 📄,📋 | ✓ | divergence-based | ✗ | BERT |
| MAUVE Pillutla et al. (2021) | 💬,🌿 | ✓ | divergence-based | ✗ | GPT-2 |
| ParaScore Shen et al. (2022) | ⇄ | ✓ | embedding sim | ✗ | RoBERTa-large |
| UniEvalDialogue Zhong et al. (2022) | 💬 | ✓ | LM regression | ✓ | T5 |
| UniEvalSum Zhong et al. (2022) | 📄,📋 | ✓ | LM regression | ✓ | T5 |
| UpdateROUGE Iv et al. (2022) | 📄 | ✗ | n-gram overlap | ✗ | N.A. |
| LENS Maddela et al. (2023) | 📄,📖 | ✓ | LM regression | ✓ | T5 |
| *Reference-Free Metrics: do not require a gold reference.* | | | | | |
| FKGL Kincaid et al. (1975) | 📄,📖 | ✗ | rule-based | ✗ | N.A. |
| Perplexity Jelinek et al. (2005) | 💬,🌿,</> | ✓ | fluency | ✗ | GPT-2 Large |
| DistinctNGrams Li et al. (2016) | 💬,🌿 | ✗ | diversity ratio | ✗ | N.A. |
| SelfBLEU Zhu et al. (2018) | 💬,🌿,⇄ | ✗ | diversity ratio | ✗ | N.A. |
| YiSi-2 Lo (2019) | 🌐 | ✓ | embedding sim | ✗ | mBERT |
| SummaQA Scialom et al. (2019) | 📄 | ✓ | LM regression | ✓ | BERT |
| FactCC Kryscinski et al. (2020) | 📄 | ✓ | LM regression | ✓ | BERT |
| Toxicity Vidgen et al. (2021) | 💬,🌿,🛡 | ✓ | classification | ✓ | RoBERTa |
| ParaScoreFree Shen et al. (2022) | ⇄,📋 | ✓ | embedding sim | ✗ | RoBERTa-large |
| Sentiment Camacho-collados et al. (2022) | 💬,🌿 | ✓ | classification | ✓ | RoBERTa |
| UniEvalFact Zhong et al. (2022) | 📄,📋 | ✓ | LM regression | ✓ | T5 |
| LENS_SALSA Heineman et al. (2023) | 📄,📖 | ✓ | LM regression | ✓ | T5 |
| FastTextEducationalValue Tsui & Nguyen (2024) | 📄,📖 | ✗ | classification | ✓ | FastText |
| FastTextNSFW Soldaini et al. (2024) | 💬,🛡 | ✗ | classification | ✓ | FastText |
| FastTextToxicity Soldaini et al. (2024) | 💬,🛡 | ✗ | classification | ✓ | FastText |
| GRMRewardModel Yang et al. (2024) | 💬,🛡 | ✓ | LM regression | ✓ | Llama-3.2-3B |
| INFORM Reward Model 70B Minghao Yang (2024) | 💬 | ✓ | LM regression | ✓ | Llama-3.1-70B |
| LDL Reward Model 27B Chen et al. (2025) | 💬,</> | ✓ | LM regression | ✓ | Gemma 2-27B |
| MathProcessRewardModel Zhang et al. (2025) | ▦,📄 | ✓ | classification | ✓ | Qwen2.5 7B |

Table 4: Comparison of generative evaluation metrics. **Icons:** 🌐 Machine Translation, 📄 Summarization, ⇄ Paraphrasing, 💬 Dialogue/Chat, 🌿 Storytelling/Creative Writing, 🖼 Image Captioning/Multimodal, 🛡 Safety/Moderation, 📋 Data-to-Text Generation, 📖 Education/Readability, </> Code Generation, ▦ Math/Problem Solving, ✎ String-Distance/Edit-Based.

**SALSA.** In researching text simplification metrics we found that an extension to the LENS (Maddela et al., 2023) metric exists which is meant to better align with human judgement. It was also a related metric to the SimpEval (Maddela et al., 2023) paper. Thus we chose to implement **SALSA** (Heineman et al., 2023) in our MetricBank as it was intended as one of the recommended "Best" metrics for our *in-distribution* SimpEval task.

| Metric | GPU | CPU | Time (ms) |
|---|---|---|---|
| INFORMRewardModel | 129.62 GB | 2.04 GB | 1041 |
| LDLRewardModel | 104.17 GB | 2.06 GB | 1921 |
| GRMRewardModel | 6.02 GB | 1.96 GB | 61 |
| UniEvalDialogue | 3.07 GB | 3.10 GB | 262 |
| UniEvalSum | 3.07 GB | 3.10 GB | 211 |
| UniEvalFact | 3.07 GB | 3.09 GB | 61 |
| Perplexity_gpt2-large | 3.00 GB | 1.47 GB | 48 |
| BLEURT | 2.15 GB | 2.75 GB | 43 |
| BARTScore_bart-large-cnn | 1.52 GB | 1.34 GB | 49 |
| SummaQA | 1.25 GB | 1.51 GB | 879 |
| YiSi | 687 MB | 1.39 GB | 35 |
| Sentiment | 485 MB | 1.36 GB | 19 |
| Toxicity | 485 MB | 1.36 GB | 39 |
| FactCC | 427 MB | 1.29 GB | 17 |
| ParaScoreFree | 346 MB | 1.68 GB | 12 428 |
| ParaScore | 338 MB | 1.05 GB | 4 543 |
| MOVERScore_distilbert-base-uncased | 262 MB | 1.50 GB | 2 899 |
| BERTScore_roberta-large | 8 MB | 1.47 GB | 1 303 |
| PRMRewardModel | 0 MB | 13.64 GB | 6 359 |
| MAUVE_max | 0 MB | 4.22 GB | 3 236 |
| FastTextEducationalValue | 0 MB | 3.73 GB | 6 |
| LENS | 0 MB | 3.25 GB | 3 408 |
| LENS_SALSA | 0 MB | 2.84 GB | 426 |
| FastTextToxicity | 0 MB | 1.67 GB | 11 |
| FastTextNSFW | 0 MB | 1.67 GB | 6 |
| InfoLM | 0 MB | 1.12 GB | 2 338 |
| METEOR | 0 MB | 1.08 GB | 27 |
| FKGL | 0 MB | 894 MB | 6 |
| TER | 0 MB | 731 MB | 26 064 |
| CHRF | 0 MB | 730 MB | 36 |
| DistinctNGram | 0 MB | 730 MB | 19 |
| iBLEU | 0 MB | 730 MB | 18 |
| BLEU | 0 MB | 729 MB | 7 |
| LevenshteinDistance_min | 0 MB | 729 MB | 0 |
| SelfBLEU | 0 MB | 729 MB | 6 |
| HammingDistance_min | 0 MB | 729 MB | 0 |
| JaroWinklerSimilarity_max | 0 MB | 729 MB | 0 |
| GLEU | 0 MB | 728 MB | 9 |
| SARI | 0 MB | 728 MB | 95 |
| JaccardDistance_min | 0 MB | 728 MB | 0 |
| CharCut | 0 MB | 728 MB | 1 237 |
| UpdateROUGE | 0 MB | 728 MB | 96 |
| NIST | 0 MB | 728 MB | 21 |
| LevenshteinRatio_max | 0 MB | 727 MB | 0 |
| JaroSimilarity_max | 0 MB | 727 MB | 0 |
| ROUGE | 0 MB | 726 MB | 487 |
| CIDEr_n4_sig6.0 | 0 MB | 726 MB | 31 |
| PseudoPARENT | 0 MB | 726 MB | 10 |

Table 5: Maximum CI upper-bound GPU/CPU memory and latency per metric.

**FastText Classifiers.** We wanted to add diversity to our MetricBank by including more classifiers for various higher-level concepts, but we didn't want to add unnecessary expenses to running

the metrics. FastText Classifiers are a nice compromise which are quick to run on CPU but also have reasonable classification accuracy. We implement **FastTextNSFW** and **FastTextToxicity** from Dolma (Soldaini et al., 2024), and we take **FastTextEducationalValue** (Tsui & Nguyen, 2024) which has been used for data filtering to attempt to find Text-Book quality training data.

## C  PROMPTS AND SIGNATURES

### C.1  LLM-AS-A-JUDGE PROMPTS

We use the LLM-as-a-Judge Prompts from the original human annotation process for a given dataset whenever available. We consider these as a strong baseline as these instructions were designed to be useful instructions for human annotators and ideally were the underlying instructions guiding their annotation decisions.

---

**Task: SimpEval**

**LLM-as-a-Judge Prompt:**

```
## Rating Sentences

The goal is to **rate sentences** by how well they **simplify the
original sentence**.

### Scoring Guidelines

| Score | When to assign it |
|-------|------------------|
| **100** | The sentence is **fully simplified**, entirely fluent,
and **preserves the core meaning** of the original. |
| **75**  | The sentence is **somewhat simpler**, mostly fluent, and
the meaning is **close** to the original. |
| **50**  | The sentence is simpler, **somewhat fluent**, and the
meaning is **similar** to the original. |
| **25**  | The sentence is equivalently simple, still has some
fluency, but **loses the meaning**. |
| **0**   | The sentence is **completely unreadable**. |

> **Most scores will lie somewhere in this range – feel free to give
specific scores (e.g., 83, 67) rather than only the five anchors.**

---

### Examples

| Score | Example Simplified Sentence | Why this score? |
|-------|----------------------------|-----------------|
| **100** | *It will then **move away from the river bed** and sink
back to the bottom to digest its food.* | Reads fluently **and**
keeps the original meaning ("it" gets unstuck, moves down, digests
food). |
| **75** | *Due to this, **a lot of mosques don't enforce these
rules** but both men and women should follow them.* | Minor fluency
issue, but meaning matches the original. |
| **0** | *A gadget javascript a is and / checking wikipedia an
snippet that can be enabled simply by , or css option in your
wikipedia preferences.* | Sentence is **unreadable**. |
```

**Task: HelpSteer2**

**LLM-as-a-Judge Prompt:**

```
**Helpfulness/Understanding:**
- 4 - The response is extremely helpful and completely aligned with
the spirit of what the prompt
was asking for.
- 3 - The response is mostly helpful and mainly aligned with what
the user was looking for, but
there is still some room for improvement.
- 2 - The response is partially helpful but misses the overall goal
of the user's query/input in some
way. The response did not fully satisfy what the user was looking
for.
- 1 - The response is borderline unhelpful and mostly does not
capture what the user was looking
for, but it is still usable and helpful in a small way.
- 0 - The response is not useful or helpful at all. The response
completely missed the essence of
what the user wanted.
```

**Task: EvalGenProduct**

**LLM-as-a-Judge Prompt:**

```
Is this response good (1) or bad (0)?
```

**Task: RealHumanEval**

**LLM-as-a-Judge Prompt:**

```
Would you accept this code edit/addition (1) or reject it (0)?
```

**Task: CoGymTravelOutcome**

**LLM-as-a-Judge Prompt:**

```
Overall rating to the final outcome (i.e., travel plan, analysis
result) (1-5 scale)

(1) "Extremely dissatisfied",
(2) "Somewhat dissatisfied",
(3) "Neutral",
(4) "Somewhat satisfied",
(5) "Extremely satisfied"
```

## C.2 MISC PROMPTS

---

### MetricCard Generation

**Prompt:**

You are an expert in natural language processing and technical
documentation, specializing in metrics for evaluating generative
models. I am building a metric bank to recommend the best metrics
for various generative tasks. Each metric in this bank will have a
corresponding Metric Card, which provides standardized, detailed
documentation about the metric. These Metric Cards will serve as a
key resource for researchers and practitioners, helping them select
the right metric for their task.

## Your Task

Using the provided materials, including the original paper,
reference implementations, the Metric Card Template, and the BLEU
Metric Card Example, your task is to draft a comprehensive Metric
Card for the given metric. The documentation must:
        1.      Follow the provided template closely, ensuring
adherence to its format and required sections.
        2.      Incorporate relevant details from the original paper
and reference materials, ensuring technical accuracy and
completeness.
        3.      Match the style and quality of the BLEU example,
which serves as an exemplar for clarity, structure, and precision.

Specific Instructions
        1.      Key Sections to Address: Ensure each required
section of the template is filled out thoughtfully and thoroughly,
including:
                -       Metric Description
                -       Inputs and Outputs
                -       Formal Definition
                -       Applicability and Limitations
                -       Known Limitations and Related Metrics
        2.      If Information is Unclear or Missing: Do not
fabricate or make assumptions. If information is unavailable,
unclear, or not included in the provided context, leave that section
blank or mark it as "Needs more information."
        3.      Markdown Formatting: Output the completed Metric
Card as a markdown text block rather than rendering or printing the
markdown directly.  This means you must surround your answer in ```.
 Also start the block with "---" as shown in the examples.  Do not
end the block with "---".
        4.      Focus on Consistency: Use the provided categorical
suggestions (see below) to ensure uniformity across all Metric
Cards, particularly in fields like "Metric Type," "Domain," and
"Tasks."
        5.      Mathematical Formatting:
                -       Use $ for inline math expressions (e.g.,
$r$, not $ r $).
                -       Use $$ for block math expressions and ensure
a full line break before and after each block math expression. This
formatting ensures proper rendering in markdown.
                -       Example of proper usage for $$:

** Correct **
```
Where:
- $CHRP$ is the average precision of character and word n-grams:

```
$$
CHRP = \frac{1}{N} \sum_{n=1}^N \frac{\text{n-grams in hypothesis
and reference}}{\text{total n-grams in hypothesis}}
$$

- $CHRR$ is the average recall of character and word n-grams:

$$
CHRR = \frac{1}{N} \sum_{n=1}^N \frac{\text{n-grams in hypothesis
and reference}}{\text{total n-grams in reference}}
$$
```

** Incorrect **
```
Where:
- $CHRP$ is the average precision of character and word n-grams:
  $$
  CHRP = \frac{1}{N} \sum_{n=1}^N \frac{\text{n-grams in hypothesis
and reference}}{\text{total n-grams in hypothesis}}
  $$
- $CHRR$ is the average recall of character and word n-grams:
  $$
  CHRR = \frac{1}{N} \sum_{n=1}^N \frac{\text{n-grams in hypothesis
and reference}}{\text{total n-grams in reference}}
  $$
```

        -        Ensure all block math expressions are clearly
separated from list items or inline text.
                -        Add a space after operators like \sum, \max,
or any LaTeX commands followed by an underscore (_) to prevent
Markdown parsers from interpreting _ as italic markers.  Mainly it
is critical to put a space before "_". For example:

** Correct **
```
$$
R _{\text{BERT}} = \frac{\sum _{x _{i} \in x} \text{idf}(x _{i})
\cdot \max _{\hat{x} _{j} \in \hat{x}} x _{i^\top} \hat{x}
_{j}}{\sum _{x _{i} \in x} \text{idf}(x _{i})}
$$
```

** Incorrect **
```
$$
R_{\text{BERT}} = \frac{\sum_{x_i \in x} \text{idf}(x_i) \cdot
\max_{\hat{x}_j \in \hat{x}} x_i^\top \hat{x}_j}{\sum_{x_i \in x}
\text{idf}(x_i)}
$$
```

        6. Citation: It is imperative that you do NOT make this up.
If the user does not explicitly provide the bibtex citation for the
metric then you must say [More Information Needed].  If a citation
is provided you must copy it EXACTLY.  Do NOT try to simplify any of
the components such as the author list with an ellipsis.

## Categorical Suggestions for Consistency

```
Note: These suggestions are not exhaustive. While you should
prioritize using the categories listed here for consistency, you may
add new categories if the metric clearly warrants them.

### Domains

These represent broad areas of application for the metric. Choose
one or more:
        -       Text Generation
        -       Speech Generation
        -       Code Generation
        -       Multimodal Generation
        -       Image Captioning
        -       Dialogue Systems
        -       Storytelling

### Tasks

These are specific tasks or use cases where the metric applies.
Choose one or more:
        -       Machine Translation
        -       Summarization
        -       Paraphrasing
        -       Data-to-Text Generation
        -       Image-to-Text Generation
        -       Dialogue Generation
        -       Style Transfer
        -       Creative Writing (e.g., poetry, storytelling)
        -       Code Completion
        -       Response Generation

### Metric Type

These classify the metric based on its design and purpose. Choose
one:
        -       Surface-Level Similarity (e.g., BLEU, ROUGE)
        -       Semantic Similarity (e.g., BERTScore)
        -       Fluency (e.g., perplexity-based metrics)
        -       Diversity (e.g., distinct-n)
        -       Robustness (e.g., adversarial robustness metrics)
        -       Fairness
        -       Faithfulness (e.g., factual consistency metrics)
        -       Reference-Free (e.g., coherence or novelty scoring)
        -       Explainability

### Inputs

These describe what the metric requires for evaluation:
        -       Reference-Based
        -       Reference-Free
        -       Input-Required
        -       Input-Optional

## Materials You Will Be Provided
        1.      Original Paper: The foundational paper introducing
or defining the metric.
        2.      Reference Implementations (when available):
Documentation from popular implementations (e.g., SacreBLEU README
for BLEU).
        3.      Metric Card Template: The standardized structure for
all Metric Cards (see below).
```

```
        4.       BLEU Metric Card Example: A high-quality example for
reference.

=== TEMPLATE FOR METRIC CARDS ===
---
# Metric Card for {{ metric_name | default("Metric Name", true) }}

{{ metric_summary | default("A brief description of the metric and
its purpose.", true) }}

## Metric Details

### Metric Description

{{ metric_description | default("Detailed explanation of the metric,
including how it is calculated and what it measures.", true) }}

- **Metric Type:** {{ metric_type | default("[More Information
Needed]", true) }}
- **Range:** {{ metric_range | default("[More Information Needed]",
true) }}
- **Higher is Better?:** {{ higher_is_better | default("[More
Information Needed]", true) }}
- **Reference-Based?:** {{ reference_based | default("[More
Information Needed]", true) }}
- **Input-Required?:** {{ input_required | default("[More
Information Needed]", true) }}

### Formal Definition

{{ metric_definition | default("Mathematical formula or detailed
algorithmic definition.", true) }}

### Inputs and Outputs

- **Inputs:**
  {{ metric_inputs | default("Description of required inputs (e.g.,
generated text, reference text, input prompt).", true) }}

- **Outputs:**
  {{ metric_outputs | default("Description of the metric output
(e.g., scalar score, distribution).", true) }}

## Intended Use

### Domains and Tasks

- **Domain:** {{ domain | default("[More Information Needed]", true)
}}
- **Tasks:** {{ tasks | default("[More Information Needed]", true) }}

### Applicability and Limitations

- **Best Suited For:** {{ best_suited_for | default("[More
Information Needed]", true) }}
- **Not Recommended For:** {{ not_recommended_for | default("[More
Information Needed]", true) }}

## Metric Implementation

### Reference Implementations
```

```
- **Libraries/Packages:** {{ libraries | default("[More Information
Needed]", true) }}

### Computational Complexity

- **Efficiency:** {{ efficiency | default("[More Information
Needed]", true) }}
- **Scalability:** {{ scalability | default("[More Information
Needed]", true) }}

## Known Limitations

{{ known_limitations | default("[More Information Needed]", true) }}

- **Biases:** {{ biases | default("Potential biases inherent in the
metric.", true) }}
- **Task Misalignment Risks:** {{ task_misalignment | default("[More
Information Needed]", true) }}
- **Failure Cases:** {{ failure_cases | default("[More Information
Needed]", true) }}

## Related Metrics

{{ related_metrics | default("[More Information Needed]", true) }}

## Further Reading

- **Papers:** {{ papers | default("[More Information Needed]", true)
}}
- **Blogs/Tutorials:** {{ blogs | default("[More Information
Needed]", true) }}

## Citation

{{ bibtex_citation | default("[More Information Needed]", true) }}

## Metric Card Authors

- **Authors:** {{ metric_authors | default("[More Information
Needed]", true) }}
- **Acknowledgment of AI Assistance:**
  {{ ai_assistance | default("Portions of this metric card were
drafted with assistance from generative AI. All content has been
reviewed and curated by the author to ensure accuracy.", true) }}
- **Contact:** {{ metric_contact | default("[More Information
Needed]", true) }}
======

=== BLEU Metric Card Example ===
---
# Metric Card for BLEU

BLEU (Bilingual Evaluation Understudy) is a widely used metric for
evaluating the quality of text generated in tasks like machine
translation and summarization. It measures the overlap of n-grams
between a generated text and one or more reference texts, with a
brevity penalty to penalize overly short translations. SacreBLEU, a
modern implementation, ensures reproducibility and standardization
of BLEU scores across research.

## Metric Details
```

```
### Metric Description

BLEU evaluates the quality of text generation by comparing n-grams
in the generated output with those in one or more reference texts.
It computes modified precision for n-grams and combines scores using
a geometric mean, with a brevity penalty to ensure the length of the
generated text matches that of the references. Higher BLEU scores
indicate closer similarity to the references.

- **Metric Type:** Surface-Level Similarity
- **Range:** 0 to 1
- **Higher is Better?:** Yes
- **Reference-Based?:** Yes
- **Input-Required?:** No

### Formal Definition

$$
\text{BLEU} = \text{BP} \cdot \exp \left( \sum_{n=1}^N w_n \log p_n
\right)
$$

where:
- $\text{BP} = \min(1, e^{1 - r/c})$ is the brevity penalty,
- $r$ is the effective reference length (based on the closest
matching reference length for each sentence),
- $c$ is the candidate translation length,
- $p_n$ is the modified precision for n-grams of length $n$,
- $w_n$ are weights for each n-gram (commonly uniform, $w_n =
\frac{1}{N}$).

### Inputs and Outputs

- **Inputs:**
  - Generated text (candidate translation)
  - Reference text(s) (gold-standard translations)

- **Outputs:**
  - Scalar BLEU score (range: 0 to 1)

## Intended Use

### Domains and Tasks

- **Domain:** Text Generation
- **Tasks:** Machine Translation, Summarization, Data-to-Text
Generation

### Applicability and Limitations

- **Best Suited For:**
  Structured tasks with a clear correspondence between generated and
reference texts, such as translation or summarization.

- **Not Recommended For:**
  Open-ended or creative generation tasks where diversity or
semantic similarity matters more than lexical overlap (e.g.,
storytelling, dialogue).

## Metric Implementation

### Reference Implementations
```

- **Libraries/Packages:**
  - [SacreBLEU](https://github.com/mjpost/sacrebleu) (robust, standard implementation)
  - [NLTK](https://www.nltk.org/api/nltk.translate.html) (basic Python implementation)
  - [Hugging Face `evaluate`](https://huggingface.co/docs/evaluate) (integrated metric framework)

### Computational Complexity

- **Efficiency:**
  BLEU is computationally efficient, requiring $O(n \cdot m)$ operations for $n$-gram matching where $n$ is the number of words in the candidate text and $m$ is the number of reference words. SacreBLEU optimizes tokenization and scoring, making it highly suitable for large-scale evaluations.

- **Scalability:**
  BLEU scales well across datasets of varying sizes due to its simple design. SacreBLEU further supports evaluation with multiple references, diverse tokenization schemes, and language-specific preprocessing, making it adaptable to diverse evaluation setups.

## Known Limitations

- **Biases:**
  - BLEU penalizes valid paraphrases or semantically equivalent outputs that do not match reference n-grams exactly.
  - The brevity penalty can overly penalize valid shorter outputs, particularly for tasks where shorter text may be acceptable or even preferred (e.g., summarization).

- **Task Misalignment Risks:**
  - BLEU is not designed for evaluating tasks with high diversity in acceptable outputs (e.g., open-ended dialogue).
  - Scores depend on the quality and number of references; fewer or inconsistent references can lead to misleading evaluations.

- **Failure Cases:**
  - BLEU struggles to capture semantic adequacy beyond lexical similarity. For instance, it cannot identify whether a translation preserves the meaning of the original sentence if word choices diverge significantly.

## Related Metrics

- **ROUGE:** Often used for summarization tasks, emphasizing recall over precision.
- **METEOR:** Incorporates synonym matching for better semantic alignment.
- **BERTScore:** Uses contextual embeddings for semantic similarity.

## Further Reading

- **Papers:**
  - [Original BLEU Paper (Papineni et al., 2002)](https://www.aclweb.org/anthology/P02-1040)
  - [SacreBLEU: A Call for Clarity in Reporting BLEU Scores (Post, 2018)](https://www.aclweb.org/anthology/W18-6319)

- **Blogs/Tutorials:**

```
  - [Understanding BLEU](https://machinelearningmastery.com
/calculate-bleu-score-for-text-python/)
  - [SacreBLEU Documentation](https://github.com/mjpost/sacrebleu)

## Citation

@inproceedings{papineni-etal-2002-bleu,
    title = "{B}leu: a Method for Automatic Evaluation of Machine
Translation",
    author = "Papineni, Kishore  and
      Roukos, Salim  and
      Ward, Todd  and
      Zhu, Wei-Jing",
    editor = "Isabelle, Pierre  and
      Charniak, Eugene  and
      Lin, Dekang",
    booktitle = "Proceedings of the 40th Annual Meeting of the
Association for Computational Linguistics",
    month = jul,
    year = "2002",
    address = "Philadelphia, Pennsylvania, USA",
    publisher = "Association for Computational Linguistics",
    url = "https://aclanthology.org/P02-1040/",
    doi = "10.3115/1073083.1073135",
    pages = "311--318"
}

## Metric Card Authors

- **Authors:** Michael J. Ryan
- **Acknowledgment of AI Assistance:**
  Portions of this metric card were drafted with assistance from
OpenAI's ChatGPT, based on user-provided inputs and relevant
documentation. All content has been reviewed and curated by the
author to ensure accuracy.
- **Contact:** michaeljryan@stanford.edu
======

The metric you will be designing a card for is {Metric Name}

=== {SUPPLEMENTAL MATERIALS} ===

======

Now please write a high quality metric card for {Metric Name} given
the provided materials!

Final **Important** Note: If the provided materials do not give
enough information about a particular point for the metric (e.g.
limitations or biases aren't listed) then do NOT make things up.
You can leave blanks or "Needs more information" where needed.  It
is absolutely essential not to make things up or guess when
producing this documentation otherwise future researchers and
engineers will be confused and led astray.  Avoid making up links
that you aren't fully confident in the url.

Remember to surround your answer in '''.  Thanks!
```

## C.3 DSPy SIGNATURES

---

**Signature: `GeneratePerturbationStrategies`**

**Instruction:**

```
You will be given:
- A Task description
- A Dimension to prioritize when perturbing outputs
- The Example Input, optional Example Reference, and Example Output

Instructions:
Your primary focus should be on degrading performance along the
    specified Dimension.
1. Begin with a rich reasoning paragraph (3-5 sentences) that
    explores a variety of ways to subtly degrade model outputs. Do
    not reference the specific example.
2. Under the heading **Strategies:**, list 1-3 numbered, high-level
    perturbation strategies.
    - Each strategy should be a short phrase (5-15 words) naming the
    category of change, followed by one concise sentence of abstract
    explanation.
    - Do not include concrete rewrites, instance-specific examples,
    or example sentences.
```

**Inputs:**

| Field | Type | Description |
|---|---|---|
| task | str | The task the model was originally trying to complete |
| example_sets | list[str] | Example inputs, outputs, and (optionally) references showing task performance |
| dimension | str | The dimension to prioritize for the perturbation |

**Outputs:**

| Field | Type | Description |
|---|---|---|
| perturbation_ strategies | list[str] | 1-3 high-level strategies to test robustness |

---

**Signature: `PerturbWorse`**

**Instruction:**

```
You will be given:
    - A Task description
    - A Dimension to prioritize when perturbing outputs
    - The Example Input, optional Example Reference, and Model Output
    - A perturbation_strength value ("subtle" or "obvious")
    - A list of perturbation_strategies to apply

Instructions:
Your goal is to apply each strategy to the Model Output and produce
    a degraded version that specifically harms performance along the
    given Dimension, using the specified strength.
Under the heading **Perturbed Outputs:**, return exactly one
    perturbed output per strategy.
    - For **subtle** strength, introduce minimal distortion.
```

```
    - For **obvious** strength, introduce more pronounced
      degradation.
Do **not** include any reasoning, explanations, or examples -- only
    the perturbed text.
```

**Inputs:**

| Field | Type | Description |
|---|---|---|
| task | str | The task that the model was originally trying to complete |
| dimension | str | The dimension to prioritize for the perturbation (this should be the aspect of the model output that is most impacted by the perturbation) |
| input | str | The input provided to the model |
| references | Union[list[str], None] | The references of good outputs (may be None) |
| model_output | str | The output produced by the model |
| perturbation_ strength | Literal['subtle', 'obvious'] | The strength of the perturbation (subtle or obvious) |
| perturbation_ strategies | list[str] | The perturbation strategies to use |

**Outputs:**

| Field | Type | Description |
|---|---|---|
| perturbed_ outputs | list[str] | Perturbed text that is worse than the original model output. Produce one perturbed output per strategy. |

**Signature: PerturbSame**

**Instruction:**

```
You will be given:
    - A Task description
    - A Dimension to preserve when perturbing outputs
    - The Example Input, optional Example Reference, and Model Output
    - A perturbation_strength value ("subtle" or "obvious")

Instructions:
Apply a perturbation to the Model Output that **maintains**
    performance on the specified Dimension.
Under the heading **Perturbed Output:** return exactly one string:
    - For **subtle** strength, apply a minimal change that does not
    impair the target Dimension.
    - For **obvious** strength, apply a more noticeable change that
    still keeps the target Dimension intact.
Some examples of types of perturbations would include: rephrasing,
    reordering, replacing words with synonyms, stylistic changes,
    etc. that do not impair the target Dimension.
If any change would harm the specified Dimension, simply return the
    original Model Output.
After producing your original plan/reasoning do **not** include any
    more reasoning, explanations, or examples -- only the perturbed
    text.
```

**Inputs:**

| Field | Type | Description |
|---|---|---|
| task | str | The task that the model was originally trying to complete |
| input | str | The input provided to the model |
| references | Union[list[str], None] | The references of good outputs (may be None) |
| model_output | str | The output produced by the model |
| perturbation_ strength | Literal['subtle', 'obvious'] | The strength of the perturbation (subtle or obvious) |
| dimension | str | The aspect of the model output that MUST be preserved in quality |

**Outputs:**

| Field | Type | Description |
|---|---|---|
| perturbed_output | str | Perturbed text that preserves performance along the given Dimension. |

**Signature: LLMAsAJudgeSignature**

**Instruction:**

```
Given an input text, the task description that the model was trying
    to follow, and a measure to rate the text on, return a score on
    this measure.
```

**Inputs:**

| Field | Type | Description |
|---|---|---|
| text | Any | The input text that we want to rate. |
| task_description | Any | A description of the task that the model was trying to solve when it generated the text. Could be left blank if not available. |
| measure | Any | The measure that we want to rate the text on. |
| suggested_range | Any | The suggested range of possible values for the measure. |

**Outputs:**

| Field | Type | Description |
|---|---|---|
| score | Any | The score that the text should receive on this measure. |

**Signature: LLMMetricRecommendationSignature**

**Instruction:**

```
I am looking for a metric to evaluate the attached task. In
    particular I care about the specific target measurement that I
    attached.
```

```
Please help me decide from among the metrics that I have attached
    documentation for which one is most relevant to the task and
    target.

Please provide a ranking of the metrics from most relevant to least
    relevant for the task and target above.
You can reason first about what makes a metric relevant for the task
    and target, and then provide your ranking.

IMPORTANT: The final ranking should be a list of EXACT metric class
    names (no hyphens, no spaces, no extra words).  Use the METRIC
    NAME not what it is called in the documentation.
For example, use "SelfBLEU" not "Self-BLEU", use "BERTScore" not
    "BERT Score", use "BLEU" not "BLEU Score".

The final ranking should just be a list of metric names, in order
    from most relevant to least relevant.
The list should be exactly `num_metrics_to_recommend` items long.
```

**Inputs:**

| Field | Type | Description |
|---|---|---|
| task_description | str | A description of the task that an LLM performed and that I now want to evaluate. |
| target | str | The specific target measurement that I want to evaluate about the task. |
| metric_ documentation | List[str] | A list of metric names and their documentation. The documentation will contain the metric name, as well as many details about the metric. |
| num_metrics_ to_recommend | int | The number of metrics to recommend. It is imperative to target this number or very very close to it. We will do more extensive filtering later. |

**Outputs:**

| Field | Type | Description |
|---|---|---|
| ranking | List[str] | A numbered list of EXACT metric class names (no hyphens, no spaces, no extra words), in order from most relevant to least relevant. The list should be of length 'num_metrics_to_recommend'. You should write the number in front of the metric name (e.g '1. METRIC1_NAME', '2. METRIC2_NAME', etc.). REMEMBER: Put quotes around EACH number + metric name pair, not just one set of quotes for the full string. IMPORTANT: Refer to "METRIC NAME: ..." for the exact name of the metric or it won't be a match. |

**Signature: GenerateRubricSignature**

**Instruction:**

```
Given a dataset, task description, and an evaluation metric,
    generate a rubric for the metric scoring from 1 to 5.
```

**Inputs:**

| Field | Type | Description |
|---|---|---|
| task_description | Any | A description of the task that the model is trying to solve. |
| good_examples | Any | A list of good examples of outputs for a model. |
| bad_examples | Any | A list of bad examples of outputs for a model. |
| metric_title | Any | The title of the metric. |
| metric_ description | Any | A description of the metric. |

**Outputs:**

| Field | Type | Description |
|---|---|---|
| score_one_ description | Any | A description of what a score of 1 means. This can be a bullet point list of what criteria to look for in assigning a score of 1. |
| score_two_ description | Any | A description of what a score of 2 means. This can be a bullet point list of what criteria to look for in assigning a score of 2. |
| score_three_ description | Any | A description of what a score of 3 means. This can be a bullet point list of what criteria to look for in assigning a score of 3. |
| score_four_ description | Any | A description of what a score of 4 means. This can be a bullet point list of what criteria to look for in assigning a score of 4. |
| score_five_ description | Any | A description of what a score of 5 means. This can be a bullet point list of what criteria to look for in assigning a score of 5. |

**Signature: GenerateAxisOfVariationSignature**

**Instruction:**

```
Given a task description, a target metric, and good/bad examples,
    generate a list of axes of variation which could be used to
    explain the differences between the good and bad examples.
    These axes of variation will be used as measures to evaluate the
    model's performance, so they should be informative and useful
    for the model to improve on.
```

**Inputs:**

| Field | Type | Description |
|---|---|---|
| `task_description` | `str` | A description of the overall task the model is trying to solve. |
| `target_name` | `Optional[str]` | Optional hint of the target metric/column we care about. Could be 'None' or something generic like 'quality' or 'score'. |
| `good_examples` | `List[str]` | A list of examples with *high* quality according to the target metric. |
| `bad_examples` | `List[str]` | A list of examples with *low* quality according to the target metric. |
| `num_axes_to _generate` | `int` | The number of axes of variation to generate. |

**Outputs:**

| Field | Type | Description |
|---|---|---|
| `axes_of _variation` | `List[str]` | An ordered list (most-important first) describing possible axes of variation. Please bold the name of the axis of variation (e.g. **Axes Name**), and ALSO include a brief sentence-long explanation of the axis of variation. (e.g. **Axes Name** Brief Explanation). Please include exactly 'num_axes_to_generate' axes of variation in the output. Avoid special characters since they sometimes mess up the parsing. |

## D EXAMPLE METRIC CARD: BLEU

**Metric Card for BLEU**

BLEU (Bilingual Evaluation Understudy) is a widely used metric for evaluating the quality of text generated in tasks like machine translation and summarization. It measures the overlap of n-grams between a generated text and one or more reference texts, with a brevity penalty to penalize overly short translations. SacreBLEU, a modern implementation, ensures reproducibility and standardization of BLEU scores across research.

**Metric Details**

*Metric Description*
BLEU evaluates the quality of text generation by comparing n-grams in the generated output with those in one or more reference texts. It computes modified precision for n-grams and combines scores using a geometric mean, with a brevity penalty to ensure the length of the generated text matches that of the references. Higher BLEU scores indicate closer similarity to the references.

- **Metric Type:** Surface-Level Similarity
- **Range:** 0 to 1
- **Higher is Better?:** Yes
- **Reference-Based?:** Yes
- **Input-Required?:** No

*Formal Definition*

$$\text{BLEU} = \text{BP} \cdot \exp\left(\sum_{n=1}^{N} w_n \log p_n\right)$$

- $\text{BP} = \min(1, e^{1-r/c})$ is the brevity penalty,
- $r$ is the effective reference length (based on the closest matching reference length for each sentence),
- $c$ is the candidate translation length,
- $p_n$ is the modified precision for n-grams of length $n$,
- $w_n$ are weights for each n-gram (commonly uniform, $w_n = \frac{1}{N}$).

*Inputs and Outputs*

- **Inputs:**
  - Generated text (candidate translation)
  - Reference text(s) (gold-standard translations)
- **Outputs:**
  - Scalar BLEU score (range: 0 to 1)

**Intended Use**

*Domains and Tasks*

- **Domain:** Text Generation
- **Tasks:** Machine Translation, Summarization, Data-to-Text Generation

*Applicability and Limitations*

- **Best Suited For:** Structured tasks with a clear correspondence between generated and reference texts, such as translation or summarization.
- **Not Recommended For:** Open-ended or creative generation tasks where diversity or semantic similarity matters more than lexical overlap (e.g., storytelling, dialogue).

**Metric Implementation**

*Reference Implementations*

- **Libraries/Packages:**
  - SacreBLEU (robust, standard implementation)
  - NLTK (basic Python implementation)
  - Hugging Face `evaluate` (integrated metric framework)

*Computational Complexity*

- **Efficiency:** BLEU is computationally efficient, requiring $O(n \cdot m)$ operations for $n$-gram matching where $n$ is the number of words in the candidate text and $m$ is the number of reference words. SacreBLEU optimizes tokenization and scoring, making it highly suitable for large-scale evaluations.
- **Scalability:** BLEU scales well across datasets of varying sizes due to its simple design. SacreBLEU further supports evaluation with multiple references, diverse tokenization schemes, and language-specific preprocessing, making it adaptable to diverse evaluation setups.

**Known Limitations**

- **Biases:**
  - BLEU penalizes valid paraphrases or semantically equivalent outputs that do not match reference n-grams exactly.
  - The brevity penalty can overly penalize valid shorter outputs, particularly for tasks where shorter text may be acceptable or even preferred (e.g., summarization).
- **Task Misalignment Risks:**
  - BLEU is not designed for evaluating tasks with high diversity in acceptable outputs (e.g., open-ended dialogue).
  - Scores depend on the quality and number of references; fewer or inconsistent references can lead to misleading evaluations.
- **Failure Cases:**
  - BLEU struggles to capture semantic adequacy beyond lexical similarity. For instance, it cannot identify whether a translation preserves the meaning of the original sentence if word choices diverge significantly.

**Related Metrics**

- **ROUGE:** Often used for summarization tasks, emphasizing recall over precision.
- **METEOR:** Incorporates synonym matching for better semantic alignment.
- **BERTScore:** Uses contextual embeddings for semantic similarity.

**Further Reading**

- **Papers:**
  - Original BLEU Paper (Papineni et al., 2002)
  - SacreBLEU: A Call for Clarity in Reporting BLEU Scores (Post, 2018)
- **Blogs/Tutorials:**
  - Understanding BLEU
  - SacreBLEU Documentation

**Metric Card Authors**

- **Authors:** Michael J. Ryan
- **Acknowledgment of AI Assistance:** Portions of this metric card were drafted with assistance from OpenAI's ChatGPT, based on user-provided inputs and relevant documentation. All content has been reviewed and curated by the author to ensure accuracy.
- **Contact:** michaeljryan@stanford.edu

# E AUTOMETRICS DESIGN ABLATIONS

## E.1 RETRIEVE

For our retrieval experiments we run all metrics in the MetricBank to get the ground truth kendall correlation on the development set. With this we know the true rank order of the metrics. We then perform retrieval using a set of retrieval algorithms, namely BM25, ColBERT, Faiss, and using an

| Method | NDCG | | | | Recall | | | |
|---|---|---|---|---|---|---|---|---|
| | @1 | @5 | @10 | @20 | @1 | @5 | @10 | @20 |
| BM25 | 0.208 ± 0.274 | 0.342 ± 0.171 | 0.427 ± 0.143 | 0.567 ± 0.16 | 0.065 ± 0.095 | 0.224 ± 0.146 | 0.418 ± 0.173 | **0.788 ± 0.319** |
| ColBERT | 0.272 ± 0.293 | 0.343 ± 0.203 | 0.442 ± 0.178 | 0.57 ± 0.174 | 0.059 ± 0.092 | 0.212 ± 0.155 | 0.441 ± 0.273 | 0.776 ± 0.361 |
| Faiss | 0.103 ± 0.059 | 0.227 ± 0.144 | 0.326 ± 0.163 | 0.461 ± 0.199 | 0.018 ± 0.058 | 0.171 ± 0.204 | 0.353 ± 0.256 | 0.694 ± 0.427 |
| LLMRec | 0.31 ± 0.334 | 0.396 ± 0.249 | 0.478 ± 0.219 | 0.602 ± 0.196 | 0.088 ± 0.101 | 0.294 ± 0.204 | 0.465 ± 0.273 | 0.641 ± 0.264 |
| BM25→LLMRec | 0.316 ± 0.323 | 0.42 ± 0.226 | 0.498 ± 0.197 | 0.603 ± 0.186 | **0.094 ± 0.101** | 0.312 ± 0.179 | 0.494 ± 0.257 | 0.665 ± 0.245 |
| ColBERT→LLMRec | **0.403 ± 0.387** | **0.462 ± 0.28** | **0.528 ± 0.238** | **0.631 ± 0.212** | **0.094 ± 0.101** | **0.329 ± 0.225** | **0.518 ± 0.266** | 0.694 ± 0.21 |
| Faiss→LLMRec | 0.164 ± 0.186 | 0.324 ± 0.234 | 0.393 ± 0.215 | 0.529 ± 0.216 | 0.065 ± 0.095 | 0.247 ± 0.226 | 0.4 ± 0.27 | 0.6 ± 0.304 |

Table 6: Average performance (± std) across all tasks/axes using Kendall ground truth (recommendations from `qwen3`).

| Method | NDCG | | | | Recall | | | |
|---|---|---|---|---|---|---|---|---|
| | @1 | @5 | @10 | @20 | @1 | @5 | @10 | @20 |
| BM25 | 0.208 ± 0.274 | 0.342 ± 0.171 | 0.427 ± 0.143 | 0.567 ± 0.16 | 0.065 ± 0.095 | 0.224 ± 0.146 | 0.418 ± 0.173 | 0.788 ± 0.319 |
| ColBERT | 0.272 ± 0.293 | 0.343 ± 0.201 | 0.42 ± 0.179 | 0.568 ± 0.167 | 0.059 ± 0.092 | 0.247 ± 0.191 | 0.429 ± 0.27 | **0.8 ± 0.338** |
| Faiss | 0.098 ± 0.059 | 0.21 ± 0.128 | 0.314 ± 0.167 | 0.461 ± 0.19 | 0.012 ± 0.048 | 0.159 ± 0.169 | 0.371 ± 0.304 | 0.729 ± 0.378 |
| LLMRec | 0.261 ± 0.302 | 0.416 ± 0.249 | 0.502 ± 0.235 | 0.585 ± 0.217 | 0.076 ± 0.099 | 0.347 ± 0.243 | 0.518 ± 0.316 | 0.759 ± 0.326 |
| BM25→LLMRec | 0.206 ± 0.197 | 0.394 ± 0.196 | 0.47 ± 0.175 | 0.576 ± 0.162 | 0.076 ± 0.099 | 0.347 ± 0.233 | 0.512 ± 0.257 | 0.794 ± 0.297 |
| ColBERT→LLMRec | **0.328 ± 0.31** | **0.475 ± 0.251** | **0.55 ± 0.214** | **0.628 ± 0.198** | **0.1 ± 0.102** | **0.388 ± 0.246** | **0.565 ± 0.301** | 0.759 ± 0.333 |
| Faiss→LLMRec | 0.157 ± 0.124 | 0.325 ± 0.205 | 0.406 ± 0.201 | 0.526 ± 0.212 | 0.065 ± 0.095 | 0.276 ± 0.226 | 0.424 ± 0.31 | 0.635 ± 0.37 |

Table 7: Average performance (± std) across all tasks/axes using Kendall ground truth (recommendations from `gpt4o-mini`).

LLM with all documents in context. We additionally try pipelined versions of all of these retrievers feeding into an LLM. We report Recall@[1,5,10,20] and NDCG@[1,5,10,20] in Table 6 for Qwen3-32B and Table 7 for GPT-4o-mini.

Overall we find that ColBERT → LLMRec is consistently the best approach for retrieval, performing the best across 14/16 of our evaluation settings. Thus, we use ColBERT → LLMRec for all metric retrieval in the main paper.

### E.2 GENERATE

We test eight different approaches to metric generation. Of these approaches five of them are cheap to produce, while three of them are expensive/time-consuming to produce.

For **CodeGen** we prompt an LLM to propose "axes of variation" from high/low examples and then synthesize small, executable Python snippets that implement a scoring function (`compute_score`). The generated code is cleaned, validated on a sample, and—if it errors—automatically repaired once by an LLM. We support both reference-free and reference-based variants.

For **G-Eval** (Liu et al., 2023) we convert each axis into a concrete evaluation criterion, auto-generate numbered evaluation steps, and prompt an LLM judge to produce a brief rationale followed by a discrete score (1–5). We request token-level log probabilities and, at the final score position (found by scanning backward), extract the logprobs over tokens $\{1, 2, 3, 4, 5\}$, softmax-normalize, and return the probability-weighted expectation $\hat{s} = \sum_{s=1}^{5} s\, P(s \mid \text{prompt}, \text{rationale})$. Both reference-free and reference-based variants are supported.

For **Single Criteria** (Saad-Falcon et al., 2024) LLM-as-a-Judge, we show high-scoring and low-scoring data points to an LLM and ask for "axes of variation." Each axis becomes its own metric, with the LLM prompted to output an integer score from 1–5.

For **Rubric** (Gunjal et al., 2025) we add an additional step to Single Criteria where we ask an LLM to generate explanations of what 1–5 scores should contain for each rubric item.

For **Rubric (Prometheus)** (Kim et al., 2024) we first synthesize a five-level rubric (descriptions for scores 1–5) from dataset examples, then use a Prometheus evaluator (e.g., M-Prometheus-14B) to assign scores conditioned on that rubric. This keeps the rubric explicit while using a strong, specialized judge.

**Finetune** is our first expensive to produce metric. For this we fine-tune a ModernBERT-Large regression head (with LoRA/PEFT) on formatted input–output (and references when available) to directly predict the target score. We use an 80/20 train/validation split, optimize with AdamW, and save the resulting adapter as a learned metric that runs without an LLM at inference.

For **Examples** we separate the provided human-rated examples into quintiles. Based on the context length of the LLM judge we determine how many examples we can reasonably sample from each quintile without exceeding the context length. We try 5 randomly sampled sets of uniformly distributed examples as context in an LLM-judge prompt and select the set that minimizes average distance to human labels on the trainset.

For **Prompt Optimization (MIPROv2)** we run DSPy's MIPROv2 (Opsahl-Ong et al., 2024) optimizer with `auto_mode="medium"` on the provided data to generate informative examples and rewrite the evaluation prompt for an LLM judge.

| Task (Measure) | Cheap to Produce | | | | | Expensive to Produce | | |
|---|---|---|---|---|---|---|---|---|
| | Code Gen | G-Eval | Single Criterion | Rubric (DSPy) | Rubric (Prometheus) | Finetune | Examples | MIPROv2 |
| *In-Distribution Tasks*: some metrics in our bank were designed to directly evaluate these tasks. | | | | | | | | |
| SummEval (coherence) | $0.098 \pm 0.019$ | $0.105 \pm 0.023$ | $\mathbf{0.194 \pm 0.010}$ | $0.173 \pm 0.017$ | $0.140 \pm 0.016$ | $0.104 \pm 0.016$ | $0.226 \pm 0.019$ | $\mathbf{0.227 \pm 0.044}$ |
| SummEval (consistency) | $0.083 \pm 0.018$ | $0.102 \pm 0.013$ | $\mathbf{0.173 \pm 0.023}$ | $0.160 \pm 0.026$ | $0.122 \pm 0.015$ | $0.095 \pm 0.042$ | $\mathbf{0.226 \pm 0.066}$ | $0.199 \pm 0.030$ |
| SummEval (fluency) | $0.057 \pm 0.016$ | $0.076 \pm 0.011$ | $\mathbf{0.121 \pm 0.009}$ | $0.110 \pm 0.015$ | $0.096 \pm 0.017$ | $0.061 \pm 0.016$ | $\mathbf{0.146 \pm 0.015}$ | $0.136 \pm 0.048$ |
| SummEval (relevance) | $0.097 \pm 0.025$ | $0.144 \pm 0.026$ | $0.213 \pm 0.017$ | $0.189 \pm 0.018$ | $0.151 \pm 0.014$ | $0.067 \pm 0.043$ | $0.243 \pm 0.022$ | $\mathbf{0.263 \pm 0.022}$ |
| Primock57 (inc_plus_omi) | $0.105 \pm 0.036$ | $0.086 \pm 0.017$ | $\mathbf{0.247 \pm 0.031}$ | $0.188 \pm 0.043$ | $0.196 \pm 0.025$ | $0.090 \pm 0.057$ | $0.253 \pm 0.057$ | $\mathbf{0.258 \pm 0.067}$ |
| Primock57 (incorrect) | $0.145 \pm 0.073$ | $0.060 \pm 0.014$ | $\mathbf{0.250 \pm 0.059}$ | $0.169 \pm 0.070$ | $0.202 \pm 0.026$ | $0.026 \pm 0.029$ | $\mathbf{0.266 \pm 0.039}$ | $0.213 \pm 0.164$ |
| Primock57 (omissions) | $0.123 \pm 0.059$ | $0.061 \pm 0.021$ | $0.119 \pm 0.029$ | $0.116 \pm 0.025$ | $\mathbf{0.129 \pm 0.020}$ | $0.125 \pm 0.077$ | $\mathbf{0.169 \pm 0.023}$ | $0.122 \pm 0.097$ |
| Primock57 (time_sec) | $0.102 \pm 0.038$ | $0.053 \pm 0.009$ | $\mathbf{0.159 \pm 0.026}$ | $0.132 \pm 0.016$ | — | $0.058 \pm 0.049$ | $0.057 \pm 0.050$ | $\mathbf{0.129 \pm 0.041}$ |
| SimpEval (score) | $0.100 \pm 0.037$ | $0.184 \pm 0.028$ | $0.229 \pm 0.019$ | $0.192 \pm 0.012$ | $0.155 \pm 0.017$ | $0.046 \pm 0.037$ | $0.216 \pm 0.036$ | $\mathbf{0.243 \pm 0.130}$ |
| SimpDA (fluency) | $0.180 \pm 0.013$ | $0.364 \pm 0.022$ | $\mathbf{0.521 \pm 0.014}$ | $0.511 \pm 0.016$ | $0.460 \pm 0.025$ | $0.050 \pm 0.051$ | $0.582 \pm 0.017$ | $\mathbf{0.583 \pm 0.058}$ |
| SimpDA (meaning) | $0.252 \pm 0.066$ | $0.397 \pm 0.030$ | $\mathbf{0.590 \pm 0.016}$ | $0.570 \pm 0.020$ | $0.546 \pm 0.026$ | $0.055 \pm 0.038$ | $\mathbf{0.632 \pm 0.025}$ | $0.625 \pm 0.024$ |
| SimpDA (simplicity) | $0.173 \pm 0.024$ | $0.305 \pm 0.033$ | $\mathbf{0.523 \pm 0.030}$ | $0.481 \pm 0.021$ | $0.442 \pm 0.015$ | $0.041 \pm 0.058$ | $0.584 \pm 0.025$ | $\mathbf{0.628 \pm 0.035}$ |
| HelpSteer (coherence) | $0.029 \pm 0.004$ | $0.162 \pm 0.027$ | $\mathbf{0.229 \pm 0.013}$ | $0.190 \pm 0.013$ | — | $0.014 \pm 0.009$ | $0.297 \pm 0.023$ | $\mathbf{0.297 \pm 0.006}$ |
| HelpSteer (complexity) | $\mathbf{0.223 \pm 0.083}$ | $0.122 \pm 0.029$ | $0.149 \pm 0.042$ | $0.184 \pm 0.035$ | — | $0.071 \pm 0.070$ | $\mathbf{0.221 \pm 0.050}$ | $0.095 \pm 0.018$ |
| HelpSteer (correctness) | $0.068 \pm 0.007$ | $0.270 \pm 0.024$ | $\mathbf{0.356 \pm 0.024}$ | $0.342 \pm 0.018$ | — | $0.044 \pm 0.027$ | $0.392 \pm 0.017$ | $\mathbf{0.424 \pm 0.009}$ |
| HelpSteer (helpfulness) | $0.066 \pm 0.019$ | $0.241 \pm 0.018$ | $\mathbf{0.333 \pm 0.011}$ | $0.327 \pm 0.018$ | — | $0.049 \pm 0.030$ | $\mathbf{0.407 \pm 0.016}$ | $0.402 \pm 0.013$ |
| HelpSteer (verbosity) | $\mathbf{0.290 \pm 0.028}$ | $0.154 \pm 0.043$ | $0.193 \pm 0.051$ | $0.252 \pm 0.053$ | — | $0.084 \pm 0.031$ | $\mathbf{0.406 \pm 0.015}$ | $0.103 \pm 0.028$ |
| HelpSteer2 (coherence) | $0.024 \pm 0.005$ | $0.116 \pm 0.019$ | $\mathbf{0.154 \pm 0.005}$ | $0.138 \pm 0.020$ | — | $0.043 \pm 0.032$ | $\mathbf{0.192 \pm 0.016}$ | $0.169 \pm 0.028$ |
| HelpSteer2 (complexity) | $\mathbf{0.113 \pm 0.040}$ | $0.074 \pm 0.024$ | $0.091 \pm 0.013$ | $0.100 \pm 0.014$ | — | $0.096 \pm 0.000$ | $\mathbf{0.335 \pm 0.074}$ | $0.065 \pm 0.045$ |
| HelpSteer2 (correctness) | $0.052 \pm 0.012$ | $0.167 \pm 0.012$ | $\mathbf{0.245 \pm 0.007}$ | $0.212 \pm 0.016$ | — | $0.037 \pm 0.035$ | $\mathbf{0.332 \pm 0.017}$ | $0.320 \pm 0.019$ |
| HelpSteer2 (helpfulness) | $0.068 \pm 0.009$ | $0.134 \pm 0.015$ | $\mathbf{0.217 \pm 0.019}$ | $0.183 \pm 0.008$ | $0.135 \pm 0.008$ | $0.026 \pm 0.015$ | $0.293 \pm 0.020$ | $\mathbf{0.309 \pm 0.015}$ |
| HelpSteer2 (verbosity) | $0.224 \pm 0.018$ | $0.161 \pm 0.031$ | $0.210 \pm 0.016$ | $\mathbf{0.234 \pm 0.048}$ | — | $0.167 \pm 0.068$ | $\mathbf{0.432 \pm 0.015}$ | $0.081 \pm 0.315$ |
| *Out-of-Distribution Tasks*: no metric is specifically designed for these – tests generalization and metric generation. | | | | | | | | |
| EvalGenProduct (grade) | $0.262 \pm 0.046$ | $0.285 \pm 0.029$ | $\mathbf{0.343 \pm 0.085}$ | $0.303 \pm 0.072$ | $0.201 \pm 0.021$ | $0.210 \pm 0.236$ | $0.145 \pm 0.046$ | $\mathbf{0.216 \pm 0.173}$ |
| EvalGenMedical (grade) | $0.262 \pm 0.046$ | $0.285 \pm 0.029$ | $\mathbf{0.343 \pm 0.085}$ | $0.303 \pm 0.072$ | $0.201 \pm 0.021$ | $0.210 \pm 0.236$ | $0.145 \pm 0.046$ | $\mathbf{0.216 \pm 0.173}$ |
| RealHumanEval (accepted) | $0.046 \pm 0.007$ | $0.039 \pm 0.013$ | $\mathbf{0.115 \pm 0.009}$ | $0.088 \pm 0.013$ | $0.079 \pm 0.011$ | $0.037 \pm 0.025$ | $0.091 \pm 0.019$ | $\mathbf{0.153 \pm 0.028}$ |
| CoGymTravelProcess (agentRating) | $\mathbf{0.208 \pm 0.027}$ | $0.185 \pm 0.061$ | $0.115 \pm 0.050$ | $0.101 \pm 0.044$ | $0.121 \pm 0.056$ | $0.218 \pm 0.246$ | $0.144 \pm 0.098$ | $0.090 \pm 0.046$ |
| CoGymTravelProcess (communicationRating) | $0.172 \pm 0.075$ | $\mathbf{0.285 \pm 0.066}$ | $0.168 \pm 0.098$ | $0.167 \pm 0.082$ | $0.165 \pm 0.028$ | $\mathbf{0.238 \pm 0.281}$ | $0.220 \pm 0.154$ | $0.180 \pm 0.195$ |
| CoGymTravelOutcome (outcomeRating) | $0.337 \pm 0.059$ | $0.318 \pm 0.100$ | $0.429 \pm 0.068$ | $\mathbf{0.448 \pm 0.117}$ | $0.413 \pm 0.057$ | $0.298 \pm 0.472$ | $\mathbf{0.558 \pm 0.131}$ | $0.518 \pm 0.273$ |
| CoGymTabularProcess (agentRating) | $0.254 \pm 0.150$ | $0.487 \pm 0.132$ | $0.538 \pm 0.067$ | $\mathbf{0.598 \pm 0.082}$ | $0.403 \pm 0.108$ | $0.475 \pm 0.203$ | $0.560 \pm 0.395$ | $\mathbf{0.637 \pm 0.315}$ |
| CoGymTabularProcess (communicationRating) | $0.360 \pm 0.046$ | $0.608 \pm 0.088$ | $0.791 \pm 0.093$ | $0.779 \pm 0.147$ | $\mathbf{0.798 \pm 0.049}$ | — | $\mathbf{0.890 \pm 0.125}$ | $0.787 \pm 0.501$ |
| CoGymTabularOutcome (outcomeRating) | $0.363 \pm 0.217$ | $0.349 \pm 0.173$ | $0.367 \pm 0.160$ | $0.201 \pm 0.081$ | $\mathbf{0.634 \pm 0.117}$ | — | $0.363 \pm 0.362$ | $0.200 \pm 0.227$ |
| **Average** | $0.159$ | $0.206$ | $0.281$ | $0.263$ | $0.276$ | $0.108$ | $0.323$ | $0.287$ |

Table 8: Metric generation performance (Kendall's Tau) with 95% confidence intervals over 5 independent runs. Each generator produces metrics using persistent train sets, then correlation with human annotations is measured on persistent validation sets. Cheap methods (left) generate 10 metrics per trial, expensive methods (right) generate 1 metric per trial (except finetune which generates 10). Results show correlation between generated metrics and ground-truth human annotations across diverse tasks using the Qwen3 32B model.

# F    ADDITIONAL EXPERIMENTS

## F.1    ROBUSTNESS FOR ALL METRICS

Here we include the results of the robustness experiment for all baseline metrics tested. We report results in Figure 7.

We find that "Best Metric" tends to be quite stable, while the LLM Based Metrics (DNAEval, LLM-Judge, and AutoMetrics) stand out on robustness.

| Task (Measure) | Cheap to Produce | | | | | Expensive to Produce | | |
| --- | --- | --- | --- | --- | --- | --- | --- | --- |
| | Code Gen | G-Eval | Single Criterion | Rubric (DSPy) | Rubric (Prometheus) | Finetune | Examples | MIPROv2 |
| ***In-Distribution Tasks:*** *some metrics in our bank were designed to directly evaluate these tasks.* | | | | | | | | |
| SimpEval (score) | 0.127 ± 0.015 | 0.279 ± 0.024 | **0.324 ± 0.026** | 0.299 ± 0.024 | 0.166 ± 0.022 | 0.046 ± 0.037 | 0.297 ± 0.041 | **0.318 ± 0.039** |
| SimpDA (fluency) | 0.135 ± 0.020 | 0.534 ± 0.016 | 0.510 ± 0.014 | **0.573 ± 0.012** | 0.460 ± 0.013 | 0.050 ± 0.051 | **0.639 ± 0.028** | 0.635 ± 0.018 |
| SimpDA (meaning) | 0.246 ± 0.028 | 0.570 ± 0.012 | 0.551 ± 0.007 | **0.601 ± 0.022** | 0.538 ± 0.014 | 0.055 ± 0.038 | **0.686 ± 0.039** | 0.643 ± 0.022 |
| SimpDA (simplicity) | 0.092 ± 0.045 | 0.500 ± 0.016 | **0.540 ± 0.005** | 0.535 ± 0.026 | 0.463 ± 0.031 | 0.041 ± 0.058 | 0.621 ± 0.039 | **0.622 ± 0.019** |
| ***Out-of-Distribution Tasks:*** *no metric is specifically designed for these – tests generalization and metric generation.* | | | | | | | | |
| EvalGenProduct (grade) | 0.190 ± 0.032 | 0.204 ± 0.063 | **0.225 ± 0.069** | 0.175 ± 0.087 | 0.189 ± 0.080 | 0.210 ± 0.236 | 0.121 ± 0.040 | **0.248 ± 0.069** |
| EvalGenMedical (grade) | 0.190 ± 0.032 | 0.204 ± 0.063 | **0.225 ± 0.069** | 0.175 ± 0.087 | 0.189 ± 0.080 | 0.210 ± 0.236 | 0.121 ± 0.040 | **0.248 ± 0.069** |
| CoGymTravelProcess (agentRating) | **0.201 ± 0.032** | 0.113 ± 0.037 | 0.092 ± 0.027 | 0.173 ± 0.041 | 0.127 ± 0.054 | 0.218 ± 0.246 | **0.223 ± 0.108** | 0.067 ± 0.041 |
| CoGymTravelProcess (communicationRating) | **0.232 ± 0.074** | 0.176 ± 0.038 | 0.070 ± 0.035 | 0.154 ± 0.083 | 0.228 ± 0.017 | 0.238 ± 0.281 | **0.312 ± 0.086** | 0.000 ± 0.000 |
| CoGymTravelOutcome (outcomeRating) | 0.286 ± 0.049 | 0.400 ± 0.103 | 0.450 ± 0.114 | **0.474 ± 0.051** | 0.412 ± 0.092 | 0.298 ± 0.472 | 0.502 ± 0.024 | **0.513 ± 0.009** |
| CoGymTabularProcess (agentRating) | 0.273 ± 0.199 | 0.555 ± 0.098 | **0.697 ± 0.139** | 0.555 ± 0.064 | 0.435 ± 0.120 | 0.475 ± 0.203 | 0.600 ± 0.000 | **0.659 ± 0.220** |
| CoGymTabularProcess (communicationRating) | 0.404 ± 0.138 | 0.853 ± 0.091 | 0.859 ± 0.093 | **0.881 ± 0.020** | 0.763 ± 0.091 | — | 0.000 ± 0.000 | **0.890 ± 0.125** |
| CoGymTabularOutcome (outcomeRating) | 0.470 ± 0.227 | 0.547 ± 0.159 | 0.565 ± 0.090 | 0.616 ± 0.120 | **0.804 ± 0.094** | — | 0.430 ± 0.268 | **0.623 ± 0.329** |
| **Average** | 0.237 | 0.411 | 0.426 | 0.434 | 0.398 | 0.184 | 0.379 | 0.456 |

Table 9: Metric generation performance (Kendall's Tau) with 95% confidence intervals over 5 independent runs. Each generator produces metrics using persistent train sets, then correlation with human annotations is measured on persistent validation sets. Cheap methods (left) generate 10 metrics per trial, expensive methods (right) generate 1 metric per trial (except finetune which generates 10). Results show correlation between generated metrics and ground-truth human annotations across diverse tasks using the GPT-4o Mini model.

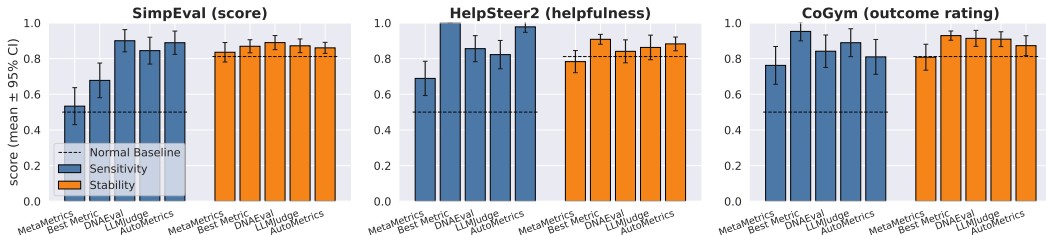

Figure 7: Sensitivity and Stability of all metrics for SimpEval, HelpSteer2, and CoGym.

## F.2 WHAT METRICS DOES AUTOMETRICS ACTUALLY SELECT?

To explore the question of what metrics AutoMetrics actually recommends we turn to the 25 trials of AutoMetrics run for our main correlation experiment from Table 2. We look exclusively at the Qwen3-32B runs. We provide a bar plot of metric types in Figure 8.

**AutoMetrics are dominated by Generated Metrics.** 103 out of the 125 total recommended metrics were automatically generated. Of the Existing metrics that were recommended 20 out of 22 were recommendations to use a reward model. This suggests that the scope of metrics to retrieve from can be dramatically reduced to primarily recommending from the generated metrics as well as a few key reward models and other model based metrics like "ParaScoreFree". This insight will in practice greatly simplify the search space for metrics and lead to a more streamlined MetricBank.

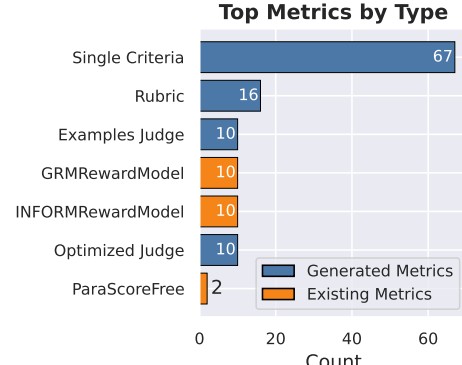

Figure 8: Breakdown of metrics recommended by AutoMetrics. Generated are most common.

## F.3 VALIDATING SENSITIVITY AND STABILITY

In order to sanity check our sensitivity and stability scores we asked a colleague not involved in our project to annotate 150 datapoints from SimpEval Maddela et al. (2023) using the original annotation rubric described in the paper. SimpEval consists of original and simple sentence pairs. We asked them to annotate 30 pairs from the original dataset, 30 pairs where the simplified sentence was perturbed in a way that does not change the quality, and 90 sentences perturbed to purposefully degrade the quality. All perturbations were following our

methodology described in 3.2. Our human annotations yielded a sensitivity of 0.8275 and stability of 0.8000 suggesting the perturbations produced the intended effect.

# G AUTOMETRICS EXAMPLES

**SimpEval — score**

**Overall Kendall $\tau$:** 0.3234

**Top 5 Metrics & Coefficients**

| Metric | Coefficient |
|---|---|
| Audience_Appropriateness_Qwen3-32B | 1.7066 |
| Conciseness_Qwen3-32B | 1.6676 |
| Readability_Score_Qwen3-32B | 1.6622 |
| Clarity_and_Readability_Rubric | 1.6345 |
| ParaScoreFree | $-1.6125$ |

**Description: Audience_Appropriateness_Qwen3-32B**

Tailors language and phrasing to suit a general audience with minimal prior knowledge of the topic.

**Description: Conciseness_Qwen3-32B**

Eliminates redundant phrases, wordiness, or tangential details while maintaining the original intent.

**Description: Readability_Score_Qwen3-32B**

Measures the text's ease of reading using standardized metrics (e.g., Flesch-Kincaid Grade Level).

**Description: `Clarity_and_Readability_Rubric`**

```
| Score | Description |
|-------|-------------|
| 1 | - The text is difficult to understand due to overly complex
     sentence structures, passive voice, or ambiguous phrasing.
- Redundant or redundant information is included, hindering
   clarity.
- Sentences are excessively long or fragmented, making it hard to
    follow the main idea.
- Jargon or technical terms are retained without simplification.
- The output fails to restructure the original sentence for
   broader accessibility. |

| 2 | - The text is somewhat clear but still contains occasional
     complex structures or passive voice.
- Some sentences are overly long or include minor redundancies.
- Ambiguity or unclear phrasing is present in parts of the output
   .
- Simplification is attempted but incomplete, leaving some
   original complexity intact.
- The main idea is generally understandable but requires effort
   to parse. |

| 3 | - The text is mostly clear, with mostly active voice and
     straightforward phrasing.
- Sentences are concise and well-structured, though a few may
   retain slight complexity.
- Minor ambiguities or redundancies are present but do not
   significantly hinder understanding.
- Simplification is effective for the core message, though some
   details may remain dense.
- The output is accessible to a general audience with minimal
   effort. |

| 4 | - The text is clear and uses active voice consistently,
    with minimal passive constructions.
- Sentences are concise, well-structured, and free of unnecessary
    complexity.
- Ambiguity is largely avoided, and phrasing is precise.
- Simplification is thorough, with original complexity reduced to
    enhance accessibility.
- The output is easy to understand for a broad audience, with
   only minor improvements possible. |

| 5 | - The text is exceptionally clear, using active voice and
    simple, direct sentence structures.
- All phrasing is unambiguous, and sentences are optimized for
   readability.
- Redundancy and complexity are entirely eliminated, with the
   core message distilled to its essentials.
- Simplification is flawless, making the content immediately
   accessible to all audiences.
- The output exemplifies best practices in clarity and
   readability, with no room for improvement. |
```

---

**Description: `ParaScoreFree`**

ParaScoreFree is a reference-free evaluation metric designed for paraphrase generation. It evaluates candidate paraphrases based on semantic similarity to the input source while encouraging lexical diversity. ParaScoreFree outputs a scalar quality score that combines BERT-based semantic similarity and normalized edit distance, offering a balance between meaning preservation and surface-level rewriting. It enables paraphrase evaluation without the need for gold reference texts, making it suitable for low-resource or open-domain settings.

---

**HelpSteer2 — helpfulness**

**Overall Kendall $\tau$:** 0.3481

**Top 5 Metrics & Coefficients**

| Metric | Coefficient |
|---|---|
| INFORMRewardModel | 0.2046 |
| HelpSteer2_helpfulness_Qwen3-32B_optimized_seed45 | 0.1853 |
| GRMRewardModel | 0.1697 |
| helpfulness_Qwen3-32B_examples | 0.1661 |
| Accuracy_and_Correctness_Qwen3-32B | 0.1625 |

---

**Description: `INFORMRewardModel`**

The INFORM Reward Model 70B (INF-ORM-Llama3.1-70B) is a large-scale outcome reward model designed to evaluate the quality of generated conversational responses. It predicts scalar reward scores for response texts, supporting preference-based fine-grained evaluations without requiring a reference response. The model is finetuned from the Llama-3.1-70B-Instruct backbone using preference-labeled datasets, employing scaled Bradley-Terry loss to incorporate preference magnitudes.

---

**Description: `HelpSteer2_helpfulness_Qwen3-32B_optimized_seed45`**

```
Given the task description, evaluation axis, input/output texts,
    and suggested score range, analyze the output text's alignment
     with the task and axis by:
1. **Assessing factual accuracy**: Verify if claims in the output
     are correct and supported by the input/text domain knowledge.
2. **Evaluating relevance**: Determine if the output addresses
    the user's intent directly, avoiding verbosity or tangential
    content.
3. **Analyzing structure and clarity**: Check if explanations are
     concise, logically organized, and accessible to the target
    audience (e.g., non-experts).
4. **Identifying gaps or errors**: Highlight missing key details,
     misinterpretations, or inaccuracies that reduce helpfulness.
5. **Scoring**: Assign a numerical score within the suggested
    range, balancing the above factors.
Use the conversation history and task description as guidance for
     context and expectations. Prioritize precision in reasoning
     and alignment with the evaluation axis.

*Showing 0 of 8 total examples.*
```

**Description: `GRMRewardModel`**

The GRMRewardModel is a general-purpose reward model designed to evaluate the quality and safety of LLM-generated outputs. It achieves high generalization performance by applying a novel regularization method on hidden states during supervised fine-tuning. GRMRewardModel is fine-tuned on the decontaminated Skywork/Skywork-Reward-Preference-80K-v0.2 dataset and achieves state-of-the-art results among models of comparable size (3B), even outperforming some 8B reward models and proprietary LLM judges on RewardBench.

**Description: `helpfulness_Qwen3-32B_examples`**

```
| Input Text | Score |
|------------|-------|
| <Input (prompt): <Can you teach me semi-definite programming in
    simple language?> Output (response): <Can you teach me how to
    use a computer in simple language?>> | 0 |
| <Input (prompt): <Delve into the nuanced benefits of engaging
   in group projects instead of the solo endeavor of individual
   projects. Craft your insights in a well-organized format,
   employing distinct headings for each category. Populate each
   section with a thoughtful list, elucidating each approach's
   merits and drawbacks. This approach aims to enhance clarity
   and the discussion's overall academ... | 0 |
| <Input (prompt): <The misery of life never appears in a clearer
    light than when a thinking person has quite plainly seen with
    horror its hazards and uncertainties and the total darkness
   in which he lives; how he cannot find anything solid, secure,
   and beyond dispute on to which he can hold; when, as I say,
   after such thoughts he does not at once destroy an existence
   that is not one, but breathi... | 1 |

*Showing 3 of 10 total examples.*
```

**Description: `Accuracy_and_Correctness_Qwen3-32B`**

The factual correctness and reliability of the information provided.

---

**EvalGenProduct — grade**

**Overall Kendall $\tau$:** 0.4178

**Top 5 Metrics & Coefficients**

| Metric | Coefficient |
|--------|-------------|
| Formatting_Compliance_Qwen3-32B | 0.1144 |
| grade_Qwen3-32B_examples | 0.1022 |
| Call_to_Action__CTA_Strength_Qwen3-32B | 0.0752 |
| Customer_Review_Integration_Rubric | 0.0747 |
| Avoidance_of_Weaknesses_Qwen3-32B | 0.0653 |

**Description: `Formatting_Compliance_Qwen3-32B`**

Good examples strictly follow Markdown structure (headers, bullet points). Bad examples include disallowed elements (links, markdown errors).

**Description: `grade_Qwen3-32B_examples`**

```
| Input Text | Score |
|------------|-------|
| <Input (Prompt): <You are an expert copywriter. You need to
    write an e-commerce product description based on the product
    details and customer reviews. Your description should be SEO-
    optimized. It should use an active voice and include the
    product's features, benefits, unique selling points without
    overpromising, and a call to action for the buyer. Benefits
    describe how product features will wor... | 0 |
| <Input (Prompt): <You are an expert copywriter. You need to
    write an e-commerce product description based on the product
    details and customer reviews. Your description should be SEO-
    optimized. It should use an active voice and include the
    product's features, benefits, unique selling points without
    overpromising, and a call to action for the buyer. Benefits
    describe how product features will wor... | 0 |
| <Input (Prompt): <You are an expert copywriter. You need to
    write an e-commerce product description based on the product
    details and customer reviews. Your description should be SEO-
    optimized. It should use an active voice and include the
    product's features, benefits, unique selling points without
    overpromising, and a call to action for the buyer. Benefits
    describe how product features will wor... | 1 |

*Showing 3 of 4 total examples.*
```

**Description: `Call_to_Action__CTA__Strength_Qwen3-32B`**

Good examples include urgent, benefit-driven CTAs (e.g., 'Order now for seasonal savings'), while bad examples have vague or missing CTAs.

**Description:** `Customer_Review_Integration_Rubric`

```
| Score | Description |
|-------|-------------|
| 1 | - **No customer reviews included** or all quotes are
    fabricated.
- Reviews are irrelevant to the product or its benefits.
- Over-cites testimonials (e.g., 5+ quotes) or includes negative
    feedback.
- Quotes are generic (e.g., "Great product!") without specific
    context. |
| 2 | - **Minimal or inconsistent use of customer reviews** (e.g
    ., 1-2 quotes).
- Quotes are vague or lack specificity (e.g., "I love this
    product!").
- Reviews may include irrelevant details or fail to align with
    the product's features/benefits.
- No clear connection between testimonials and the product's
    unique selling points. |
| 3 | - **Moderate use of customer reviews** (e.g., 2-3 quotes).
- Some quotes are specific and relevant (e.g., "This product
    works well for dry skin").
- May include 1-2 generic or slightly over-cited testimonials.
- Reviews are integrated but do not strongly enhance the
    description's persuasiveness. |
| 4 | - **Effective use of 1-2 authentic, contextually relevant
    quotes**.
- Testimonials highlight specific benefits (e.g., "The
    lightweight formula makes it perfect for travel").
- Quotes are concise, avoid over-citing, and align with the
    product's features.
- Reviews are integrated naturally into the description without
    overwhelming the reader. |
| 5 | - **Excellent integration of 1-2 highly specific, authentic
     testimonials**.
- Quotes directly tie to the product's unique selling points (e.g
    ., "The smudge-proof formula lasts all day").
- Reviews are concise, impactful, and enhance the description's
    credibility.
- No fabricated, irrelevant, or over-cited quotes; testimonials
    feel organic and persuasive. |

*Showing 3 of 4 total examples.*
```

**Description:** `Avoidance_of_Weaknesses_Qwen3-32B`

Good examples omit product drawbacks. Bad examples inadvertently mention flaws (e.g., 'may clog pores') or use hedging language.

**RealHumanEval — accepted**

**Overall Kendall $\tau$:** 0.1487

**Top 5 Metrics & Coefficients**

| Metric | Coefficient |
|---|---|
| GRMRewardModel | 0.0325 |
| INFORMRewardModel | 0.0293 |
| Code_Readability_Qwen3-32B | 0.0283 |
| RealHumanEval_accepted_Qwen3-32B_optimized_seed44 | 0.0234 |
| Modularity_and_Reusability_Qwen3-32B | 0.0218 |

### Description: `GRMRewardModel`

The GRMRewardModel is a general-purpose reward model designed to evaluate the quality and safety of LLM-generated outputs. It achieves high generalization performance by applying a novel regularization method on hidden states during supervised fine-tuning. GRMRewardModel is fine-tuned on the decontaminated Skywork/Skywork-Reward-Preference-80K-v0.2 dataset and achieves state-of-the-art results among models of comparable size (3B), even outperforming some 8B reward models and proprietary LLM judges on RewardBench.

### Description: `INFORMRewardModel`

The INFORM Reward Model 70B (INF-ORM-Llama3.1-70B) is a large-scale outcome reward model designed to evaluate the quality of generated conversational responses. It predicts scalar reward scores for response texts, supporting preference-based fine-grained evaluations without requiring a reference response. The model is finetuned from the Llama-3.1-70B-Instruct backbone using preference-labeled datasets, employing scaled Bradley-Terry loss to incorporate preference magnitudes.

### Description: `Code_Readability_Qwen3-32B`

Clarity of variable names, structure, and comments for maintainability.

### Description: `RealHumanEval_accepted_Qwen3-32B_optimized_seed44`

```
You are an expert Python code reviewer in a high-stakes software
    engineering environment where code correctness directly
    impacts mission-critical systems (e.g., financial transactions
    , medical devices, or autonomous vehicles). Your task is to
    evaluate the AI-generated code output for **absolute
    correctness** and **completeness** along the specified
    evaluation axis. A single error could lead to catastrophic
    failures. Analyze the code with extreme rigor, checking for:
1. **Logical correctness** (does it solve the task as described?)
2. **Syntax validity** (Python 3 compliance, no placeholders like
    'xrange()' or 'raw\_input()')
3. **Edge case handling** (negative numbers, empty inputs, etc.)
4. **Mathematical/statistical rigor** (valid algorithms, no
    arbitrary values like 'b = 8')
5. **Functionality** (working return statements, no stubs or
    incomplete logic).
Assign a score between 0.0 and 1.0, where 0.0 means the code is
    non-functional or completely ignores the task, and 1.0
    represents a flawless implementation. Use the input/output
    text and conversation history for context.

*Showing 0 of 8 total examples.*
```

---

**Description: `Modularity_and_Reusability_Qwen3-32B`**

Code organization into reusable functions/methods with clear separation of concerns.

---

**CoGymTravelOutcome — outcomeRating**

**Overall Kendall $\tau$:** 0.4301

**Top 5 Metrics & Coefficients**

| Metric | Coefficient |
| --- | --- |
| Cultural_and_Local_Integration_Rubric | 0.1963 |
| Cultural_and_Local_Experiences_Qwen3-32B | 0.1927 |
| Accommodation_Options_Qwen3-32B | 0.1824 |
| outcomeRating_Qwen3-32B_examples | 0.1674 |
| Feasibility_and_Realism_Qwen3-32B | 0.1620 |

**Description: `Cultural_and_Local_Integration_Rubric`**

```
| Score | Description |
|-------|-------------|
| 1 | - **Score 1 (Poor):**
- No mention of unique local experiences or cultural highlights.
- No authentic food/dining recommendations.
- Generic or irrelevant suggestions (e.g., luxury dining for a
    budget trip).
- Fails to address the user's query or intent. |
| 2 | - **Score 2 (Weak):**
- Minimal mention of local experiences (e.g., 1-2 generic
    activities like "visiting a market").
- Vague food/dining suggestions (e.g., "try local cuisine"
    without specifics).
- Lacks integration of cultural or seasonal traditions (e.g., no
    mention of KFC for Christmas).
- Missing links or references to local resources. |
| 3 | - **Score 3 (Fair):**
- Includes 1-2 specific local experiences (e.g., visiting
    Jigokudani Monkey Park).
- Mentions 1-2 authentic food/dining options (e.g., "try miso
    ramen").
- Some cultural or seasonal references (e.g., "KFC is popular for
     Christmas").
- Limited use of links or resources to support recommendations. |
| 4 | - **Score 4 (Good):**
- Includes 3-4 unique local experiences (e.g., snow monkeys,
    winter illuminations, regional festivals).
- Highlights 2-3 specific, culturally significant food/dining
    options (e.g., "try KFC for Christmas," "visit a local ramen
    shop").
- Integrates cultural/seasonal traditions (e.g., "Christmas
    markets in Hokkaido").
- Provides 1-2 links to local events, businesses, or resources. |
| 5 | - **Score 5 (Excellent):**
- Includes 5+ unique, deeply integrated local experiences (e.g.,
    snow monkeys, winter illuminations, regional festivals, and
    lesser-known gems).
- Highlights 3+ specific, culturally significant food/dining
    options with detailed descriptions (e.g., "try miso ramen at [
    specific shop]").
- Seamlessly integrates cultural/seasonal traditions (e.g., "KFC
    for Christmas, winter illuminations in Sapporo").
- Provides 3+ links to local events, businesses, or resources (e.
    g., [Sapporo White Illumination](url)).
- Tailors recommendations to the user's preferences and avoids
    irrelevant suggestions. |
```

**Description: `Cultural_and_Local_Experiences_Qwen3-32B`**

Incorporates authentic local traditions, cuisine, and unique regional activities.

**Description: `Accommodation_Options_Qwen3-32B`**

Recommends diverse lodging choices (e.g., luxury hotels, budget hostels, or ryokans).

**Description: `outcomeRating_Qwen3-32B_examples`**

```
| Input Text | Score |
|------------|-------|
| <Input (query): <I want to go to Hong Kong and spend less money
   > Output (outcome): <**Luxury Entertainment**  **Fine Dining
   :**  *  Michelin-starred restaurants: Hong Kong boasts a wide
    array of Michelin-starred restaurants offering diverse
   cuisines. Examples include Caprice (French), Lung King Hee (
   Cantonese), and Amber (Contemporary). *  Private dining
   experiences: Many high-end restaurants offer private dining
   rooms for a more... | 1.0 |
| <Input (query): <Kauai> Output (outcome): <\# Kauai Itinerary:
   5 Days of Scenery \& Bro Time (Grad Student Budget)  **Trip
   Overview:** This itinerary focuses on experiencing Kauai's
   natural beauty while keeping costs down. It's designed for a
   group of bros looking for a relaxing and scenic getaway in
   late March.  **Weather Considerations:** March is the tail end
    of the rainy season in Kauai. Expe... | 2.0 |
| <Input (query): <I will go to yangzhou city in China> Output (
   outcome): <Thank you for sharing your interests in history and
    food! Could you please let me know how many days you plan to
   stay in Yangzhou? This will help me create a well-structured
   itinerary for your trip.>> | 2.0 |

*Showing 3 of 9 total examples.*
```

**Description: `Feasibility_and_Realism_Qwen3-32B`**

Ensures the itinerary is realistic in terms of time, distance, and activity density.

---

**TauBench — reward**

**Overall Kendall $\tau$:** 0.2469

**Top 3 Metrics & Coefficients**

| Metric | Coefficient |
|--------|-------------|
| Membership_Benefit_Application_Rubric | 0.0803 |
| Escalation_Appropriateness_Rubric | 0.0599 |
| Policy_Compliance_Qwen3-32B | 0.0567 |

**Description: `Membership Benefit Application Rubric`**

```
| Score | Description |
|-------|-------------|
| 1 | - **Score 1 (Fails to apply rules)**:
- Incorrectly assigns free baggage allowances regardless of
    membership tier or cabin class.
- Applies insurance benefits to users who do not meet eligibility
     criteria (e.g., no insurance, basic economy).
- Offers compensation certificates to users who are not eligible
     (e.g., regular members without insurance).
- Fails to enforce policy restrictions (e.g., allowing basic
    economy cancellations outside the 24-hour window without
    insurance). |
| 2 | - **Score 2 (Major errors in application)**:
- Applies baggage allowances inconsistently (e.g., correct for
    some tiers but not others).
- Misapplies insurance eligibility (e.g., allows refunds for
    cancellations without valid reasons).
- Offers compensation certificates in most cases but misses key
    eligibility criteria (e.g., ignores membership tier).
- Occasionally transfers to human agents unnecessarily due to
    incorrect benefit application. |
| 3 | - **Score 3 (Partial adherence with minor errors)**:
- Correctly applies baggage allowances for most membership tiers
    but has occasional errors (e.g., miscalculates free bags for
    gold members).
- Applies insurance eligibility in most cases but fails in edge
    cases (e.g., business class cancellations without checking
    insurance status).
- Offers compensation certificates in most eligible scenarios but
     occasionally misses conditions (e.g., delayed flights without
     verifying membership).
- Rarely transfers to human agents due to minor benefit
    application issues. |
| 4 | - **Score 4 (High adherence with rare errors)**:
- Correctly applies baggage allowances for all membership tiers
    and cabin classes in most cases.
- Applies insurance eligibility accurately in nearly all
    scenarios.
- Offers compensation certificates in all eligible cases but has
    one minor oversight (e.g., miscalculating certificate amounts
    for multi-passenger reservations).
- Transfers to human agents only when necessary and for valid
    reasons. |
| 5 | - **Score 5 (Perfect adherence)**:
- Always assigns free baggage allowances correctly based on
    membership tier and cabin class.
- Applies insurance eligibility and compensation rules flawlessly
    , adhering strictly to policy.
- Never offers ineligible benefits (e.g., no certificates to
    regular members without insurance).
- Transfers to human agents only when the request falls outside
    the scope of membership benefits. |
```

**Description: `Escalation_Appropriateness_Rubric`**

```
| Score | Description |
|-------|-------------|
| 1 | - **Fails to transfer** in all cases where policy limits
    are reached or exceptions are needed.
- **Incorrectly handles** requests that require human
    intervention (e.g., proceeds with booking/canceling flights
    outside policy).
- **No adherence** to the rule of transferring for policy
    violations or exceptions. |
| 2 | - **Transfers inconsistently** (e.g., transfers in some
    policy-violating cases but not others).
- **Fails to transfer** for critical exceptions (e.g., basic
    economy cancellations without insurance, destination changes).
- **Attempts to resolve** issues beyond its scope (e.g.,
    modifying flight destinations, waiving fees without human
    input). |
| 3 | - **Transfers** in most policy-violating cases (e.g.,
    denies basic economy cancellations and transfers to human
    agents).
- **Partially handles exceptions** (e.g., transfers for
    compensation requests but not for all policy violations).
- **Some errors** in determining when to escalate (e.g.,
    transfers unnecessarily for minor issues). |
| 4 | - **Consistently transfers** when policy limits are reached
     (e.g., denies basic economy cancellations, blocks destination
     changes).
- **Transfers for exceptions** (e.g., user insists on refunds for
     non-refundable tickets, requests compensation for delays).
- **Minimal errors** in escalation decisions, with clear
    adherence to policy boundaries. |
| 5 | - **Perfectly transfers** in all required cases (e.g.,
    policy violations, exceptions, ambiguous requests).
- **Never attempts to handle** requests outside its scope (e.g.,
    denies basic economy cancellations, blocks invalid
    modifications).
- **Proactively transfers** when user intent is unclear or
    requires human judgment (e.g., personal emergencies,
    compensation negotiations). |
```

**Description: `Policy_Compliance_Qwen3-32B`**

Adherence to airline rules (e.g., no basic economy cancellations without insurance or 24-hour window).

