# OpenReview forum: "AutoMetrics: Approximate Human Judgments with Automatically Generated Evaluators"
_ICLR.cc/2026/Conference — ICLR 2026 Poster_

### Official Review · Reviewer_8L96 · 2025-10-29

**Soundness:** 3
**Presentation:** 2
**Contribution:** 3
**Rating:** 6
**Confidence:** 3

**Summary:**

The paper proposes AutoMetrics, a framework for LLM-as-a-judge that automatically generates task-specific evaluation criteria and dynamically assign weights to the criteria to form a scalar evaluation score. The framework consists of 4 steps: Generate, retrieve, regress and report. The paper conducts experiments on 5 benchmarks to show improvement in Kendall correlation with human judgement, compared with standard LLM-as-a-judge approaches.

**Strengths:**

The paper resolves a critical problem in prior works for LLM evaluation that rubrics need to be manually designed. Instead, the proposed framework automatically generates and selects criteria for specific tasks. The paper conducts sufficient experiments to support its claim.

**Weaknesses:**

Though the framework is claimed as "automatic", it still contains human-curated components. For instance, the "Retrieve" step exploits MetricBank, a static metrics collection drawn from prior literature. Therefore, it remains in question whether the generated metrics are suitable for all tasks.

Paragraph is incomplete from line 304-315, which seriously hinders content understanding.

**Questions:**

No additional questions.

---

> ### Author Response · Authors · 2025-11-27
> **Reply to Reviewer 8L96**
>
> Thank you so much for your time spent reading and providing feedback on our work\!  We are excited that you find our work resolves “a critical problem” and that “the paper conducts sufficient experiments to support its claim”.  We will address each of your weaknesses below:
>
> > W1: Though the framework is claimed as "automatic", it still contains human-curated components. For instance, the "Retrieve" step exploits MetricBank, a static metrics collection drawn from prior literature. Therefore, it remains in question whether the generated metrics are suitable for all tasks.
>
> MetricBank is intended as a resource for the community that contributors can continue to expand with metrics.  Ideally this can be a living resource that enables the community to work on better evaluators without manual labor for every user.  However, in Table 3 we also benchmark AutoMetrics using just the automatically generated metrics (no manual curation of MetricBank) and we find on most tasks this is also quite highly correlated.  This suggests that the generated metrics can be useful with or without the retrieved metrics.  As we discuss with reviewer bvUH, **the MetricBank augments and enhances the generated metrics, especially when more data is available, but the generated metrics alone can also be quite strong.**
>
> > W2: Paragraph is incomplete from line 304-315, which seriously hinders content understanding.
>
> Thank you for catching this typographical error.  This was the intended paragraph:
>
> **AutoMetrics correlates better than all baselines across all five tasks.** We find that AutoMetrics outperforms all other existing baselines on all five tasks.  While the best performing baseline is both inconsistent on dataset (LLM Judge on SimpEval, HelpSteer, EvalGen; DnA Eval on RealHumanEval and CoGym) and on the underlying model used (Existing Metrics outperform GPT-4o-mini but not Qwen3-32B). In contrast, AutoMetrics is consistently the best option regardless of dataset or underlying model.  On all datasets besides HelpSteer and CoGym, the AutoMetrics performance exceeds all baselines by greater than the 95% confidence interval. In general, AutoMetrics is the best choice for higher correlation with human ratings.
>
> We have updated this in the attached draft.

---

### Official Review · Reviewer_5zyv · 2025-11-02

**Soundness:** 3
**Presentation:** 3
**Contribution:** 3
**Rating:** 4
**Confidence:** 3

**Summary:**

The paper proposes a system that automatically creates and learns evaluation metrics for LLM-generated outputs when you only have a small amount of human feedback. It does this by using LLMs to generate candidate evaluators like rubrics and judges, and then statistically combining the best ones so that their scores align closely with real human judgments.

The fundamental problem this approach addresses is the situation where you don’t have a reward signal to optimize models: data scarcity, limited human judgments, and no ability to run full-scale preference data collection. These automatically generated metrics can then act as a proxy reward model of sorts. In other words, it’s essentially a framework for **operationalizing limited human judgments.**

In terms of validity, the authors calibrate their learned metrics to human data using Kendall’s tau correlation, so the system is empirically optimized to fit whatever limited judgments are available. It also performs well on internal coherence tests, scoring high on both sensitivity and stability, meaning it penalizes degraded outputs and stays consistent under irrelevant variation, at least within the evaluated tasks. However, this evaluation setup is circular in logic: the system’s sensitivity and stability are tested using examples generated by other LLMs, so it is being validated by the same kind of model it’s built from. These tests therefore measure LLM-to-LLM agreement rather than genuine alignment with human reasoning.

But the claimed validity of this approach is itself limited in a more fundamental way. The authors demonstrate that their generated metrics align with human judgments, but those judgments come from a small, potentially unrepresentative sample of annotators or users. In other words, the system’s validity is only as sound as the data it’s validated against. If the underlying human feedback lacks breadth, consistency, or reliability, then demonstrating correlation with it doesn’t establish true validity: it just shows agreement with a narrow and possibly biased slice of human opinion.

Despite this limitation in validity, it is my opinion that the system still demonstrates a useful way to operationalize limited human judgments.

**Strengths:**

The reframing of evaluation under limited data as a learning problem is really the "aha" moment of this paper. Operationalizing limited human data into a compositional and interpretable set of metrics is useful. The paper validates its approach across 5 domains, and provides convincing ablations of each component's contribution. Relatively well-written and accessible.

**Weaknesses:**

The main weakness is in its grounding of validity. The authors demonstrate strong correlations with small human-labeled datasets, but those datasets themselves are limited, noisy, and unrepresentative, which undermines the strength of their claims. Demonstrating correlation with a weak or narrow gold standard does not establish that AutoMetrics captures genuine human judgment, only that it fits that small sample relatively well; but whether this is truly *useful* remains an open question that is not demonstrated. Overfitting the metrics on this small data sample is the aim of the game here. Fundamentally, the missing validation is if human practitioners actually find these automated metrics meaningful or actionable in practice. The interpretability claim also remains untested- the nature of the approach itself is not sufficient evidence of it being "interpretable".

**Questions:**

NA

---

> ### Author Response · Authors · 2025-11-27
> **Reply to Reviewer 5zyv (1/2)**
>
> Thanks for your time spent reviewing our paper and your useful feedback\!  We are appreciative of your kind words about our paper’s “aha” moment of operationalizing limited human data.   We are glad that you found our ablations convincing and that you found the paper to be “well-written and accessible”.
>
> We will address each of the weaknesses you raise:
>
> > W1: this evaluation setup is circular in logic: the system’s sensitivity and stability are tested using examples generated by other LLMs, so it is being validated by the same kind of model it’s built from
>
> This is a reasonable concern so to test that the “sensitivity/stability” robustness check was operating on high quality data we recruited a colleague who was not involved in this project to help with some limited data annotation.  We replicated the SimpEval evaluation setting and asked our human annotator to rate sentences from 1-100 based on the annotation guidelines from the paper.  We asked them to rate 30 original sentences, 30 sentences we had Qwen3-32B perturb but retain quality/meaning, and 90 sentences we had Qwen3-32B perturb to purposefully reduce the quality.  We report the sensitivity/stability reported for our AutoMetrics and our human judge below:
>
> | Metric | Stability | Sensitivity |
> |:-------|-----------------:|------------------:|
> | Autometrics Regression | 0.860123 | 0.888889 |
> | human score | 0.827500 | 0.800000 |
>
> This is much better than random chance for sensitivity (0.5) and suggests that the perturbations are properly noticeable by a human judge as well.  **We will add this detail of the sensitivity/stability validation to the appendix of the paper.**
>
> > W2: The authors demonstrate strong correlations with small human-labeled datasets, but those datasets themselves are limited, noisy, and unrepresentative, which undermines the strength of their claims.
>
> This is a very important consideration for any practitioner working on evaluating their application.  Indeed this is a fundamental challenge in human evaluation as a whole.  The metrics we can uncover are only as good as the human data that is input to the framework.  Since we leverage LLMs to generate metrics which have broader knowledge of the things that might matter towards evaluation, in practice we do get fairly high level metrics that are broadly useful (we will discuss more in the next section), however we don’t wish to make any claims that AutoMetrics will generalize beyond the provided data.  In fact, fitting to the provided data can actually be a strength when it comes to settings like personalization.
>
> We don’t want users to misinterpret AutoMetrics as a panacea that can replace collecting some real diverse human data as an important evaluation step.  To avoid this **we will add the following statement to the limitations of the paper**:  AutoMetrics may only generalize as far as the provided data enables it.  Collecting real, diverse human data is still an essential part of evaluation.  The more representative and generalizable the input data is, the better and more general the AutoMetrics will be.  Users should collect data that is representative of the opinions and population that they want their evaluation to cover.

---

> > ### Author Response · Authors · 2025-11-27
> > **Reply to Reviewer 5zyv (2/2)**
> >
> > > W3: the missing validation is if human practitioners actually find these automated metrics meaningful or actionable in practice
> >
> > We really appreciate this suggestion.  Alongside the feedback from reviewer raWk we conduct a quick user study.  For your convenience the discussion of this is replicated below.
> >
> > We sent our AutoMetrics library to a few AI application developers that we know in order to collect feedback.  So far two have responded and given quite positive feedback.  The first (P1) was working on identifying what makes a good public comment on public policy/regulation. The other (P2) was working on a travel planning application similar to our CoGym task in the paper.  We summarize the key insights below:
> >
> > **Both participants found AutoMetrics connected with evaluations that experts had pointed out for their task**.  When asked about how well aligned the metrics were with expert suggestions the participants both rated 4 out of 5: Pretty close, most of the metrics were aligned with things we (or experts) found important.  P1 wrote “in our task, subject matter experts had told us to be on the lookout for some things — for example, ‘contains cost-benefit analysis’ was a prominent feature they knew to exist in good comments. To our surprise, AutoMetrics discovered this as a feature\!”  P2 noted “‘Accommodation’ ‘feasibility and realism’ actually matches the aspects that we designed in the original eval.”
> >
> > **Participants found the metrics to be insightful and useful.**  Both participants rated that they had learned something about the nature of their task from using AutoMetrics.  P1 indicated that they would use “More than 5 metrics” from autometrics (in particular this participant found the larger pool of generated metrics insightful) and P2 indicated that they would use “3 metrics” from AutoMetrics for their evaluation.  P2 explained that “I actually find the report functioning as another expert perspective on how to evaluate my task. Even though I may not use its metrics directly, it prompts me to rethink whether my evaluation makes sense.”  P1 indicated that they plan to continue using AutoMetrics for their project and both participants indicated that they would consider using AutoMetrics for their next project.
> >
> > Because this user study was quite informal and such a low sample size we don’t intend to add these results as a part of the main paper however if we get a few more participants to respond we would like to add this as additional support in the appendix.  Thanks for this useful feedback\!

---

### Official Review · Reviewer_raWk · 2025-11-03

**Soundness:** 3
**Presentation:** 2
**Contribution:** 3
**Rating:** 4
**Confidence:** 4

**Summary:**

The paper proposes AutoMetrics, a novel framework for automatically constructing evaluation metrics that approximate human judgments in subjective, open-ended tasks with scarce human feedback data. AutoMetrics consists of the four steps: generation of task-specific criteria (both from LLM-generated and existing rubrics), retrieval of top-k metrics, regression with PLS equations between the top-k metrics and gold human labels, and reports of metrics for interpretability. AutoMetrics achieves a higher correlation with human preferences than existing baselines, with only 80 human examples.

**Strengths:**

- The paper was quite easy to read and understand, with good clarity of method explanations.
- Timely topic for LLM-guided evaluations for subjective tasks with scarce human feedback data available. It was highly interesting to show that AutoMetrics can be a good alternative as a reward model for agent optimization with reduced cost and complexity of LLM alignment.

**Weaknesses:**

- The regression step fits PLS over a relatively large set of highly correlated candidate metrics after top-k retrieval. While AutoMetrics achieved a higher Kendall correlation with human annotations over baselines (Table 2), the noisy or negative tau values on the smallest tasks (notably CoGym) may indicate a known problem of spurious correlation in high-p, low-n settings induced by too many weak predictors. I acknowledge the claim that < 100 labels needed is true, but it needs more clarification to prevent a "conditionally true" statement. It seems that Autometrics could work when (1) generated, task-specific metrics are the majority among the top-k ones or (2) the task objective is close to existing metrics in MetricBank, but degrades when the top-k contains many loosely related metrics.

- AutoMetrics consists of multiple LLM-generated evaluators, thus highly likely to contain inherent biases. The paper does not run the standard robustness test, such as learning the metric on model A's outputs and evaluating on model B's outputs. Some analysis of a simple cross-model generalization table (e.g., generating/regressing using GPT-4o-generated metrics, testing on Qwen outputs) can make the claim of a "human-aligned" (after regression) approach of automatic evaluator selection processes.

**Questions:**

- As a suggestion for a stability metric, I would recommend a metric of RBO (rank-biased overlap) to measure a correlation between the rankings of AutoMetrics and human labels. RBO is top-weighted and allows unequal-length lists, a fairer metric than Kendall's tau. If you want to examine “do different runs of AutoMetrics pick roughly the same top metrics?”, RBO gives a more interpretable measure of set-level stability of the head of the ranking. Kendall's tau is a measure of pairwise ordering over the entire list, which can look bad even when the top 5 are identical.

- Generally, the paper lacks a dedicated section on limitations.

- (Not Required but Strongly Recommended as additional analysis to boost your claim) Did you run any informal user study where human users actually used these "reports" from AutoMetrics (step 4) or any adoption rate? Even a brief description would help justify this part of the contribution, measuring the actual implication/impact of AutoMetrics.

---

> ### Author Response · Authors · 2025-11-27
> **Reply to Reviewer raWk (1/3)**
>
> Thank you so much for your time reading our paper and insightful comments for improving the manuscript\!  We are pleased that you found the paper “easy to read and understand” and our case study on using AutoMetrics as a reward signal “highly interesting”.  We will address your weaknesses and questions individually:
>
> > W1: I acknowledge the claim that \< 100 labels needed is true, but it needs more clarification to prevent a "conditionally true" statement. It seems that Autometrics could work when (1) generated, task-specific metrics are the majority among the top-k ones or (2) the task objective is close to existing metrics in MetricBank, but degrades when the top-k contains many loosely related metrics.
>
> We agree with this argument and find it is aligned with our results.  Indeed on our smaller datasets (CoGym and EvalGen) we find that “Generated-Only” metrics outperform using the “Full MetricBank” (Table 3).  Also when artificially constraining the size of the RealHumanEval dataset we find at smaller scales the generated metrics are better than retrieving from the bank (Figure 4).  This leads us to enforcing this setting in AutoMetrics ‘Since most tasks will be out of distribution by nature, we default to using “Generated Only” when the user provides less than 80 training samples’ (L418-419).  This is part of what enables AutoMetrics to work well at varying scales of data without succumbing to this spurious correlation in the high-p, low-n setting.  We will clarify the writing when we discuss this in the paragraph “On out-of-distribution datasets “Generated Only” can outperform “Full” with low-resources.” (L414)
>
> **Concretely we will add this text to the paragraph**:
>
> We argue this is a product of the high-p, low-n problem in regression where having too many weak predictors and not enough data points can lead to spurious correlations.  By limiting to “generated metrics only” for low-n settings we enforce the use of stronger predictor signals.
>
> > W2: Some analysis of a simple cross-model generalization table (e.g., generating/regressing using GPT-4o-generated metrics, testing on Qwen outputs) can make the claim of a "human-aligned" (after regression) approach of automatic evaluator selection processes.
>
> This is a really useful suggestion.  There are two directions worth discussing here:
>
> 1\. One way to approach this is to evaluate system outputs of a different model using our AutoMetrics.  **We are already doing this\!**  The individual datasets come with outputs from a wide variety of AI models, and in fact very few of them come from GPT4o-mini or Qwen3-32B.
>
> 2\. Another approach is generating and regressing on metrics from one model, while deploying Autometrics with another. Something to note here is that as a part of our metric generation process we run prompt optimization on the model (for the “Optimized” condition).  Additionally prior work has shown that few-shot examples don’t always transfer between models \[1\] which impacts our “Examples” condition.  Since our metrics are generated to work with a particular model we expect a drop in performance when deploying to another model, but it remains interesting to test.  We report results from this transfer experiment below as Kendall Tau originally on that model, then transferred to the other model for both Qwen3-32B and GPT4o mini.
>
> | Dataset | Qwen3-32B original | Qwen3-32B → GPT-4o mini | GPT-4o mini original | GPT-4o mini → Qwen3-32B |
> |--------|--------------------|--------------------------|-----------------------|---------------------------|
> | SimpEval\_score | 0.311 ± 0.041 | 0.358 ± 0.074 | 0.312 ± 0.048 | 0.278 ± 0.012 |
> | HelpSteer2\_helpfulness | 0.342 ± 0.005 | 0.329 ± 0.009 | 0.324 ± 0.010 | 0.220 ± 0.015 |
> | EvalGenProduct\_grade | 0.382 ± 0.053 | 0.261 ± 0.182 | 0.316 ± 0.123 | \-0.014 ± 0.254 |
> | RealHumanEval\_accepted | 0.145 ± 0.005 | 0.155 ± 0.011 | 0.160 ± 0.000 | 0.118 ± 0.041 |
> | CoGymTravelOutcome\_outcomeRating | 0.365 ± 0.079 | 0.263 ± 0.098 | \-0.043 ± 0.082 | 0.134 ± 0.237 |
>
> **Overall AutoMetrics generally works still when transferring to new models and it even improves performance in 3 settings** (Qwen3-32B → GPT-4o mini for SimpEval and RealHumanEval // GPT-4o mini → Qwen3-32B for CoGym).  There is only one setting where it performs particularly poorly (GPT-4o mini → Qwen3-32B EvalGen).  We appreciate the suggestion to check this and feel this lends some support to the robustness of the metrics.

---

> ### Author Response · Authors · 2025-11-27
> **Reply to Reviewer raWK (2/3)**
>
> > Q1: I would recommend a metric of RBO (rank-biased overlap) to measure a correlation between the rankings of AutoMetrics and human labels. RBO is top-weighted and allows unequal-length lists, a fairer metric than Kendall's tau. If you want to examine “do different runs of AutoMetrics pick roughly the same top metrics?”
>
> This is a great suggestion\!  We test RBO in two ways, first we report it as a replacement for Kendall’s Tau when comparing our metric scores to human judgments.  These results are reported for Qwen3-32B below.
>
> | type         | SimpEval            | HelpSteer2            | EvalGen                | RealHumanEval        | CoGym                  |
> |--------------|---------------------|------------------------|-------------------------|-----------------------|-------------------------|
> | metametrics  | 0.0825 ± 0.0006     | 0.0115 ± 0.0020        | 0.2658 ± 0.0054         | 0.0089 ± 0.0014       | 0.2222 ± 0.0007         |
> | dna_eval     | 0.0428 ± 0.0122     | 0.0147 ± 0.0026        | 0.3641 ± 0.0402         | **0.0141 ± 0.0045**   | 0.4362 ± 0.0838         |
> | llm_judge    | 0.1155 ± 0.0120     | 0.0117 ± 0.0009        | 0.3692 ± 0.0220         | 0.0140 ± 0.0024       | 0.4251 ± 0.0314         |
> | autometrics  | **0.1717 ± 0.0470** | **0.0160 ± 0.0024**    | **0.3993 ± 0.0191**     | 0.0062 ± 0.0009       | **0.4558 ± 0.0610**     |
>
> AutoMetrics beats all the baselines for Qwen3-32B besides on RealHumanEval.  However, RBO is heavily determined by the size of the original set (EvalGen and CoGym are highest because of smallest dataset sizes).
>
> Perhaps more in line with the intended use of RBO we also test the stability of our metrics over multiple bootstrapped subsets of the human data.  This is in response to reviewer bvUH’s comments about stability as well.
>
> We computed an initial ranking of the metrics using the weights from the regression on all the training data.  Then we ran 2000 trials using just 80% of the data and recomputed the regression on this subset.  We compute RBO (p=0.9) between the regression ordering of each trial and the original and obtain bootstrapped confidence intervals from these 2000 trials.  We present the results for Qwen3-32B below:
>
> | Dataset               | RBO (mean ± CI) |
> |-----------------------|------------------|
> | **CoGymTravelOutcome** | 0.633 ± 0.077 |
> | **EvalGenProduct**     | 0.662 ± 0.056 |
> | **HelpSteer2**         | 0.929 ± 0.023 |
> | **RealHumanEval**      | 0.894 ± 0.032 |
> | **SimpEval**           | 0.879 ± 0.056 |
>
> As you can see **for the larger datasets the regressions are quite stable (RBO \> 0.8) and for smaller datasets (CoGym and EvalGen) the RBO is still modest (\>0.5).**  We appreciate the suggestion to run this analysis and believe it strengthens confidence in the stability of our method.
>
> > Q2: Generally, the paper lacks a dedicated section on limitations.
>
> This is a great point.  **We have drafted the following limitations section**:
>
> As a part of the AutoMetrics framework we construct and optimize metrics with particular LLMs.  Because the metric generation process involves optimizing to a particular model we have found that producing metrics with one model and running them with another reduces performance.  This suggests that when better models are released it will be important to reoptimize automatic metrics using AutoMetrics rather than just swap out the underlying LLM.
>
> AutoMetrics may only generalize as far as the provided data enables it.  Collecting real, diverse human data is still an essential part of evaluation.  The more representative and generalizable the input data is, the better and more general the AutoMetrics will be.  Users should collect data that is representative of the opinions and population that they want their evaluation to cover.
>
> AutoMetrics depends on running a regression for many predictors on a limited number of data points.  Although we took this into account with the design of our Regression step, it is still possible to run into a high-P low-N regression problem that risks spurious correlations.  To counteract accidental misuse of AutoMetrics leading to poor evaluation, we add warnings to the metric reports when the significance of the correlation with human judgments of the recommended metric is low ($p \> 0.05$).
>
> Finally, as a part of this work we do not conduct a formal user study to demonstrate the adoption of AutoMetrics among practitioners.  We have collected positive feedback on the metrics through informal tests with AI developers. We hope that by releasing and open sourcing this library, we will have the opportunity to work with the community to improve AutoMetrics.

---

> ### Author Response · Authors · 2025-11-27
> **Reply to Reviewer raWk (3/3)**
>
> > Q3: Did you run any informal user study where human users actually used these "reports" from AutoMetrics (step 4\) or any adoption rate? Even a brief description would help justify this part of the contribution, measuring the actual implication/impact of AutoMetrics.
>
> This is an excellent suggestion.  We sent our AutoMetrics library to a few AI application developers that we know in order to collect feedback.  So far two have responded and given quite positive feedback.  The first (P1) was working on identifying what makes a good public comment on public policy/regulation. The other (P2) was working on a travel planning application similar to our CoGym task in the paper.  We summarize the key insights below:
>
> **Both participants found AutoMetrics connected with evaluations that experts had pointed out for their task**.  When asked about how well aligned the metrics were with expert suggestions the participants both rated 4 out of 5: Pretty close, most of the metrics were aligned with things we (or experts) found important.  P1 wrote “in our task, subject matter experts had told us to be on the lookout for some things — for example, ‘contains cost-benefit analysis’ was a prominent feature they knew to exist in good comments. To our surprise, AutoMetrics discovered this as a feature\!”  P2 noted “‘Accommodation’ ‘feasibility and realism’ actually matches the aspects that we designed in the original eval.”
>
> **Participants found the metrics to be insightful and useful.**  Both participants rated that they had learned something about the nature of their task from using AutoMetrics.  P1 indicated that they would use “More than 5 metrics” from autometrics (in particular this participant found the larger pool of generated metrics insightful) and P2 indicated that they would use “3 metrics” from AutoMetrics for their evaluation.  P2 explained that “I actually find the report functioning as another expert perspective on how to evaluate my task. Even though I may not use its metrics directly, it prompts me to rethink whether my evaluation makes sense.”  P1 indicated that they plan to continue using AutoMetrics for their project and both participants indicated that they would consider using AutoMetrics for their next project.
>
> Because this user study was quite informal and such a low sample size we don’t intend to add these results as a part of the main paper however if we get a few more participants to respond we would like to add this as additional support in the appendix.  Thanks for this useful feedback\!
>
> **References**
>
> \[1\] Yao Lu, Max Bartolo, Alastair Moore, Sebastian Riedel, and Pontus Stenetorp. 2022. Fantastically Ordered Prompts and Where to Find Them: Overcoming Few-Shot Prompt Order Sensitivity. In Proceedings of the 60th Annual Meeting of the Association for Computational Linguistics (Volume 1: Long Papers), pages 8086–8098, Dublin, Ireland. Association for Computational Linguistics.

---

### Official Review · Reviewer_bvUH · 2025-11-04

**Soundness:** 3
**Presentation:** 2
**Contribution:** 2
**Rating:** 6
**Confidence:** 3

**Summary:**

The paper introduces AutoMetrics, a framework for building task-specific evaluation metrics from fewer than 100 human labels. It generates new LLM-based metrics, retrieves existing ones from MetricBank, and combines them via PLS regression into a composite metric that best predicts human feedback. Results across five text-generation tasks show moderate improvements in Kendall’s $\tau$ over baselines (LLM-as-a-Judge, existing metrics), and the paper introduces “Sensitivity/Stability" tests and a $\tau$-Bench proxy-reward case study.

**Strengths:**

1. The pipeline is clearly described and practically implementable.

2. The construct validity probes (Sensitivity/Stability) are a useful addition to standard correlation metrics.

3. The low-data claim (performance saturating at 80–100 samples) is empirically supported and relevant.

4. The $\tau$-Bench case study is interesting, showing the learned metric can act as a reward signal.

**Weaknesses:**

1. On out-of-distribution tasks, “Generated-only” metrics perform as well as or better than the full hybrid version, suggesting the Retrieve step (MetricBank) adds little value.

2. The PLS regression and metric generation both depend on the same small set of labels, and I am not sure if this is the principled approach (compared to split the data half and half).

3. The AutoMetrics pipeline seems more complex and computationally expensive than the baselines it's compared against.

**Questions:**

1. Given the small-N (N=80) and large-P (k=30) nature of the initial regression, how stable are the selected metrics? If you were to bootstrap the 80 human-labeled samples, would the top 5 selected metrics and their weights vary dramatically?

2. What does the failure case look like (e.g. GPT4o in CoGym)?

---

> ### Author Response · Authors · 2025-11-27
> **Reply to Reviewer bvUH (1/2)**
>
> Thank you for the time and effort spent reading our paper, we greatly appreciate your feedback and suggestions\!  We are glad that you found the method to be clear and our extensions to standard metric evaluations to be a useful contribution\!  We will address your weaknesses and questions one at a time:
>
> > W1: On out-of-distribution tasks, “Generated-only” metrics perform as well as or better than the full hybrid version, suggesting the Retrieve step (MetricBank) adds little value.
>
> There are two important considerations to note here:
>
> 1. In its ideal form the MetricBank can serve as a growing collection of the best community contributed metrics to the open source library, or for a particular user serve as a collection of internally implemented metrics known to be useful.  Thus in its current form the MetricBank may not be as important as generated metrics, yet **the definition of out-of-distribution tasks is highly dependent on the growth of the MetricBank which is an artifact that can evolve with the library**.
> 2. On the RealHumanEval out-of-distribution task we actually find a significant importance (p\<0.01) to the existing metrics.  **We find from our data scaling experiments that on out-of-distribution tasks with less data generated metrics are important but as you scale the data size AutoMetrics uncovers the existing metrics that are also useful**.  We address this finding in the paragraph “On out-of-distribution datasets \`\`Generated Only" can outperform \`\`Full" with low-resources.” (L414)
>
> > W2: The PLS regression and metric generation both depend on the same small set of labels, and I am not sure if this is the principled approach (compared to split the data half and half).
>
> This is an important consideration.  We ultimately decided to avoid subdividing our data further due to the small scale of some of our datasets (N=72 for CoGym and N=100 for EvalGen) with the expectation that users of AutoMetrics will have similarly small dataset sizes.  However we recognize at larger data scales this might be a tradeoff worth making for even more robust metric generation.  **Thus, we updated our library to allow users to specify a different “metric generation” and “regression” data split** or default to using the same data for each.
>
> > W3: The AutoMetrics pipeline seems more complex and computationally expensive than the baselines it's compared against.
>
> This is not fully accurate, although the claim certainly has merit.  There are either human or computational costs traded off in the baselines as well:
>
> 1. **Best Existing Metric**: To actually run this in practice one would need to run all metrics on a validation or train split of the dataset and pick the best one. This is a higher cost to the evaluation step of autometrics (since it isn’t filtered by retrieval).
> 2. **MetaMetrics** (Winata et al., ICLR 2025\) \[1\]: This method is computationally equivalent to the evaluation and regression steps of AutoMetrics.  Similar to the “Best Existing Metric” it is not filtered by retrieval so depending on the size of your metric bank it will be a higher cost than autometrics.
> 3. **FineTuned LLM:** This baseline incurs a higher computational resource cost of requiring compute capable of finetuning a LLM to do evaluation.  It also demands more data to be effective.
> 4. **LLM Judge**: This is computationally much cheaper than running AutoMetrics, but assumes a hidden human cost of prompt-engineering an effective LLM-as-a-Judge prompt and writing detailed guidelines (we compare against the actual annotator instructions as a prompt)
> 5. **DnA Eval** (Li et al, Coling 2025\) \[2\]**:** This incurs less of a setup (“training”) cost than AutoMetrics, but actually involves making several calls to LLMs at inference time to write new criteria for evaluation and reweight that criteria, so the cost of DnA Eval can outpace AutoMetrics at inference time when actually using the metrics.

---

> > ### Author Response · Authors · 2025-11-27
> > **Reply to Reviewer bvUH (2/2)**
> >
> > > Q1: Given the small-N (N=80) and large-P (k=30) nature of the initial regression, how stable are the selected metrics? If you were to bootstrap the 80 human-labeled samples, would the top 5 selected metrics and their weights vary dramatically?
> >
> > This is a great question\!  Combining this with the suggestion from reviewer raWk we decided to evaluate this with Rank-Biased Overlap (RBO).  We computed an initial ranking of the metrics using the weights from the regression on all the training data.  Then we ran 2000 trials using just 80% of the data and recomputed the regression on this subset.  We compute RBO (p=0.9) between the regression ordering of each trial and the original and obtain bootstrapped confidence intervals from these 2000 trials.  We present the results for Qwen3-32B below:
> >
> > | Dataset               | RBO (mean ± CI) |
> > |-----------------------|------------------|
> > | **CoGymTravelOutcome** | 0.633 ± 0.077 |
> > | **EvalGenProduct**     | 0.662 ± 0.056 |
> > | **HelpSteer2**         | 0.929 ± 0.023 |
> > | **RealHumanEval**      | 0.894 ± 0.032 |
> > | **SimpEval**           | 0.879 ± 0.056 |
> >
> > As you can see **for the larger datasets the regressions are quite stable (RBO \> 0.8) and for smaller datasets (CoGym and EvalGen) the RBO is still modest (\>0.5).**  We appreciate the suggestion to run this analysis and believe it strengthens confidence in the stability of our method.
> >
> > > Q2: What does the failure case look like (e.g. GPT4o in CoGym)?
> >
> > From observing GPT4o mini versus Qwen3-32B we find that GPT4o mini is far more generous with higher scores.  In this failure case on CoGym we identify that GPT4o mini assigns a score of 4 or 5 to all but 3 travel plans for several generated metrics.  This means there isn’t enough variance in the features to fit a good regression.  Hence the final regression is quite noisy and leads to low correlation on the test set.
> >
> > It is worth noting that this problem is detectable for practitioners using autometrics.  Typically the p-value on our correlations is quite low (average p=0.0299) indicating strong significance.  Even Qwen on Cogym has a modest significance (p=0.148 on average).  However in this failure case the p-value for GPT4o mini is vastly higher (p=0.694 on average).  **To address this we have added a warning in the report for users to carefully check their metrics if the p-value is \>0.05, and we have put an explicit warning that autometrics did not appropriately identify metrics for your task if p \> 0.3.**
> >
> > References:
> >
> > \[1\] Winata, G. I., Anugraha, D., Susanto, L., Kuwanto, G., & Wijaya, D. T. (2025). MetaMetrics: Calibrating metrics for generation tasks using human preferences. In Proceedings of the Thirteenth International Conference on Learning Representations. https://openreview.net/forum?id=slO3xTt4CG
> >
> > \[2\] Minzhi Li, Zhengyuan Liu, Shumin Deng, Shafiq Joty, Nancy Chen, and Min-Yen Kan. 2025\. DnA-Eval: Enhancing Large Language Model Evaluation through Decomposition and Aggregation. In Proceedings of the 31st International Conference on Computational Linguistics, pages 2277–2290, Abu Dhabi, UAE. Association for Computational Linguistics.

---

### Author Response · Authors · 2025-12-03
**General Response and Discussion Summary**

We thank all the reviewers for their time and effort reviewing the paper\! We believe that by incorporating the feedback provided by each reviewer we have significantly improved the manuscript and regret that we did not have the opportunity to discuss the paper further.

In order to assist our new area chair with more context we have chosen to summarize the discussion below into three sections: **Primary Strengths**, **Primary Weaknesses and how we addressed them**, and **Changes to paper/library**.

## **Primary Strengths**

### Methodology

Multiple reviewers highlighted the practicality, novelty, and problem relevance of the proposed method.

* “The pipeline is clearly described and practically implementable.” (bvUH)
* “the reframing of evaluation under limited data as a learning problem is really the ‘aha’ moment of this paper” (5zyv)
* the framework “resolves a critical problem in prior works for LLM evaluation that rubrics need to be manually designed” (8L96)

### Experiments

Reviewers consistently noted the strength and breadth of the empirical evaluation.

* “The low-data claim (performance saturating at 80–100 samples) is empirically supported and relevant.” (bvUH)
* “The paper validates its approach across 5 domains, and provides convincing ablations.” (5zyv)
* “The paper conducts sufficient experiments to support its claim.” (8L96)
* “The $\\tau$-Bench case study is interesting, showing the learned metric can act as a reward signal.” (bvUH)

### Writing and Clarity

Several reviewers found the paper clear, readable, and well organized.

* “The paper was quite easy to read and understand, with good clarity of method explanations.” (raWk)
* “Relatively well-written and accessible.” (5zyv)
* “The pipeline is clearly described and practically implementable.”(bvUH)

## **Primary Weaknesses and how we addressed them**

**NOTE: We address every weakness in the full discussion below, but primarily highlight the shared weaknesses between reviewers here**.

> how stable are the selected metrics? (bvUH, raWK)

To tackle this problem we used the Rank-Biased Overlap (RBO) metric recommended by reviewer raWK.  For each of the five datasets that we tested on we generate/compute all metrics on 5 independent seeds.  We computed an initial ranking of the metrics for each seed using the weights from the regression on all the training data. Then we ran 2000 trials using a regression on just subsets of 80% of the data. We compute RBO for these bootstrapped subsets versus the initial ranking.  This gives us a sense of if the recommended metrics would be the same over subsets of the dataset. We include detailed results in the discussion below, but our main finding is that **for the larger datasets the regressions are quite stable (RBO \> 0.8) and for smaller datasets (CoGym and EvalGen) the RBO is still modest (\>0.5).**  This enhances the confidence in the stability of our method.

> (Not Required but Strongly Recommended as additional analysis to boost your claim) Did you run any informal user study where human users actually used these "reports" from AutoMetrics (step 4\) or any adoption rate? Even a brief description would help justify this part of the contribution, measuring the actual implication/impact of AutoMetrics. (raWK, 5zyv)

We ran a small scale user study by reaching out to AI developers during the rebuttal period (aside from our own internal dogfooding of the method).  Full discussion is included below but the main takeaways are: (1) **Participants found AutoMetrics connected with evaluations that experts had pointed out for their task**.  Participants indicated that the metrics matched their own evaluations or criteria that experts had suggested.  (2) **Participants found the metrics to be insightful and useful**.  Participants indicated that they would use 3 or more of the metrics generated by AutoMetrics.

## **Changes to the paper/library**

We made the following changes to the paper and library as a result of our discussions.

1. We updated our library to allow users to specify a different “metric generation” and “regression” data split (5zyv)
2. We added a limitations section to the paper (5zyv, bvUH, raWK)
3. We added a warning in the metric report for users to carefully check their metrics if the p-value of the correlation is \>0.05, and we have put an explicit warning that autometrics did not appropriately identify metrics for your task if p \> 0.3. (bvUH)
4. We fixed the typographical error cutting off the paragraph on lines 304-315 (8L96)
5. We added the Appendix F.3 with the human validation of the sensitivity/stability scores (5zyv)

Overall we greatly appreciate all the time and effort all of our reviewers spent improving our paper\!

---

### Meta-Review · Area_Chair_16Hz · 2026-01-07

**Summary:**

Reviewers agreed the paper targets important problem of deriving task specific evaluation metrics from scarce human labels. The decision relevant concerns were about validity and robustness:

1. With ~80 labels and many candidate metrics, spurious correlations are possible. It wasn't clear whether the selected top metrics are stable.
2. On out-of-distribution or low-resource settings, "Generated-Only" sometimes matched or beat "Full," raising the question of how much MetricBank contributes.
3. Lack of a standard robustness test (learn on model A outputs, evaluate on model B outputs) and concerns that sensitivity/stability checks might reflect LLM to LLM agreement more than human reasoning.
4. The framework may overfit a narrow/limited human gold set.

**Reviewer Concerns:**

1. Authors address the first concern using a bootstrap stability analysis.
2. Authors explain the failure mode and add a default “Generated-Only” mode for <80 examples to reduce weak predictors in low-N settings, plus allow users to split metric-generation vs regression data in the library.
3. Authors provided a transfer table showing mixed but often reasonable transfer, with one notably poor setting.
4. They added human validation of sensitivity/stability on a small subset, suggesting perturbations are detectable by humans and the tests are not purely LLM self-agreement.

Still outstanding / partially addressed

1. Authors acknowledge dependence on the human sample. Agreement remains only partially resolvable without broader human studies. User study remains small/informal.
2. Authors justify MetricBank as a growing resource and show cases where existing metrics matter, but the net marginal benefit is still dataset-dependent, especially in low-resource OOD settings.

**Reviewer Scores:**

I expect borderline reviewers to move slightly upward due to the added stability analyses, cross-model transfer results, and clarified limitations, while reviewers already at 6 would likely keep their scores unchanged.

---

### Decision · Program_Chairs · 2026-01-26

Accept (Poster)